# Metric Space Magnitude for Evaluating the Diversity of Latent Representations

**Katharina Limbeck**[1,2]  **Rayna Andreeva**[3]  **Rik Sarkar**[3]  **Bastian Rieck**[1,2,4]

[1]Helmholtz Munich
[2]Technical University of Munich
[3]University of Edinburgh
[4]University of Fribourg

## Abstract

The *magnitude* of a metric space is a novel invariant that provides a measure of the 'effective size' of a space across multiple scales, while also capturing numerous geometrical properties, such as curvature, density, or entropy. We develop a family of magnitude-based measures of the intrinsic diversity of latent representations, formalising a novel notion of dissimilarity between magnitude functions of finite metric spaces. Our measures are provably stable under perturbations of the data, can be efficiently calculated, and enable a rigorous multi-scale characterisation and comparison of latent representations. We show their utility and superior performance across different domains and tasks, including (i) the automated estimation of diversity, (ii) the detection of mode collapse, and (iii) the evaluation of generative models for text, image, and graph data.

## 1   Introduction

Diversity is a key concept in representation learning, referring to the relative abundance and distinctiveness of model outputs. Given the inherent complexity of deep learning models, the evaluation of diversity is thus crucial, enabling (i) the assessment of the *intrinsic richness* of latent representations, and (ii) the evaluation of the extent to which models are capable of *preserving* the properties of an input distribution. While the quantitative evaluation of generative models in particular relies on assessing trade-offs between fidelity and diversity with regards to a known reference distribution, reference-free diversity measures are becoming increasingly relevant when a ground-truth distribution is unknown or intractable. However, reference-based diversity metrics such as *recall* are notoriously fallible, sensitive to parameter choices and therefore prone to incorrectly approximate the true data manifold, whereas reference-free diversity measures often rely on simple mean summaries that fail to pass basic consistency checks [13]. Thus, existing methods lack expressivity to fully capture what it means for a space to be diverse, resulting in a critical need for novel measures that are (i) theoretically motivated, (ii) robust to noise, and (iii) capable of encoding the intrinsic diversity of data across varying levels of similarity rather than at a single fixed threshold.

**Our contributions.** Addressing this need, we propose a novel family of diversity measures based on *metric space magnitude*, a mathematical invariant that captures numerous important multi-scale geometric characteristics of metric spaces, including curvature, density, and entropy of an input space. Metric space magnitude merely requires a notion of dissimilarity between data points, permitting it to operate on both *local* and *global* scales. Hence, magnitude is poised to compare latent spaces, yielding a compact holistic summary of diversity that satisfies relevant theoretical requirements. Our work is the first to (i) introduce magnitude as a general tool for evaluating the diversity of latent representations, and (ii) formalise a notion of difference between the magnitude of two spaces across

38th Conference on Neural Information Processing Systems (NeurIPS 2024).

multiple scales of similarity. We demonstrate that magnitude is stable and can detect curvature, highlighting its use as a multi-scale summary of the local and global geometry of data. Moreover, we empirically showcase the utility of our magnitude-based diversity measure across different modalities, namely text, image, and graph embeddings, for which we observe that our measure outperforms alternative embedding-based measures of intrinsic diversity. Finally, when a reference distribution is known, our magnitude-based notion of difference reliably detects *mode collapse* and *mode dropping*, thus assisting practitioners in model evaluation and selection.

> **In a nutshell:** We propose novel *multi-scale diversity measures* based on the *magnitude* of latent representations and show their theoretical and empirical advantages for *evaluating* the diversity of text, image, and graph *embeddings arising from generative models*.

## 2   Related Work

Latent representations and embeddings have become indispensable tools for analysing data types such as images, text, and graphs. As evidenced by LLMs, understanding semantic relationships in data requires meaningful embeddings. Our work focuses on improving representation-based diversity evaluation and we thus consider the role diversity plays in this context.

**Diversity measures.**   Assessing generative model diversity remains a challenge irrespective of the domain [39], as ground truth reference distributions or labelled data are often unavailable, and human evaluation remains costly. Thus, there exists a need for interpretable, automated and unsupervised measures of intrinsic diversity. *Reference-free evaluation* is of particular importance for assessing generated text given the black-box-nature of LLMs [6], but also applicable across modalities. Motivated by this, a varied collection of diversity measures has been proposed, many of which are task-, domain- or model-specific [13]; only a fraction of them are applicable to analysing latent representations specifically. The most flexible methods summarise intrinsic diversity using average pairwise dissimilarities like $L^p$ distances or BERT-scores [38]. More recently, Friedman and Dieng [13] proposed the Vendi Score, inspired by principles from theoretical ecology. Other diversity measures are computed directly on embedding spaces, using e.g. the geometric mean of the standard deviation across each embedding dimension [20] or cluster-based measures [10]. However, as we explore in Section 3.1, none of these measures satisfy all theoretical guarantees required by an axiomatic approach to diversity, and they are limited in expressivity, providing only snapshots of diversity at a single fixed resolution. *Reference-based metrics* define diversity as the extent to which generated samples cover the full variability of the real data [28]. Examples include the Fréchet Inception Distance (FID) or the Inception score (IS). However, they do not exclusively measure diversity but are also concerned with evaluating fidelity, i.e. the assessment of similarity between generated data and real data, making it unclear how single-number summaries such as FID and IS account for each aspect in the trade-off between diversity and quality. Thus, *precision* and *recall* have been suggested as more informative summary metrics [32] and seen various improvements [19, 28, 35]. Unfortunately, as Naeem et al. [28] show, even the improved versions of precision and recall fail to satisfy the useful conditions for strong evaluation metrics, such as (i) detecting identical reference and generated distributions, (ii) capturing mode dropping, and (iii) simplicity in selecting hyperparameters. To address these concerns, *density* and *coverage* have been proposed [28]. Nevertheless, these metrics still rely on fixed-scale manifold approximations to assess diversity making them sensitive to parameter choices. By contrast, our magnitude-based measures have less stringent assumptions and can be defined in a parameter-free fashion.

**Magnitude in machine learning.**   Since its introduction to measure biological diversity [36], magnitude was formalised by Leinster [21]. Nevertheless, despite strong geometric properties [22], magnitude has only rarely been applied in a machine learning context. Recent publications started to bridge this gap, linking magnitude to boundary detection [5], edge detection in images [1], and the generalisation error of neural networks [3], as well as demonstrating its utility for multi-objective optimisation [18]. However, the full potential of magnitude for measuring diversity remains largely unexplored since existing works ignore the nature of magnitude as an intrinsic multi-scale summary, which captures both local and global geometry and diversity of the data manifold. Our work is thus the first to leverage magnitude as a flexible, multi-scale measure of diversity in latent representations.

# 3 Methods

We first discuss the theoretical properties a suitable diversity measure should satisfy and then introduce metric space magnitude. Based on this, we outline our proposed method using magnitude for measuring the diversity of latent representation and its practical implementation.

## 3.1 Desiderata for Diversity Measures

Given the difficulty in defining diversity, diversity metrics never measure diversity itself, but rather quantify related ideas. Entropy-based approaches, including magnitude, in particular share close links to diversity, often favoured in ecology for their computational benefits and agreement with fundamental axioms of diversity [8]. Following this axiomatic approach, we highlight the following key requirements [22]:

- *Effective size:* A dataset with a fixed number of points is more diverse when points are separated e.g. distributed uniformly or maximally disordered and becomes less diverse as observations cluster together. Diversity is maximised when points are completely distinct and minimised when all observations are identical.
- *Monotonicity in observations*: Including a new observation does not decrease diversity.
- *Twin property*: Including a duplicate observation does not change diversity.
- *Multi-scale*: Diversity is summarised across multiple scales of (dis)similarity and thus captures both local and global trends in the data manifold.

This list is not conclusive; Appendix C.3 provides a more rigorous discussion of desirable properties and their definitions. We observe that many diversity measures for evaluating representations in ML do not satisfy these requirements as shown via counterexamples in Appendix C.4. For example, average similarity (AVGSIM), the most frequently-used diversity measure in ML, cannot capture nuances in diversity and fails even in simple toy scenarios [13]. Specifically, it does not give a measure of effective size and does not encode the entropy or disorder of a space, which is a key aspect of diversity. Consequently, AVGSIM fails to distinguish that a more clustered representation is less diverse than a more uniformly sampled space as illustrated in Appendix C.5.1. Likewise, the geometric mean of the standard deviations across each embedding dimension [20, GMSTDS] does *not* measure effective size, and even worse it equals zero whenever an embedding feature is constant. Even the Vendi Score [13, VS], a more purpose-built diversity measure, calculated as the exponential of the Shannon entropy of the eigenvalues of a normalised similarity matrix, shows undesirable behaviour under the inclusion of observations. Moreover, neither one of the aforementioned diversity measures fulfil the twin property nor monotonicity in observations [22], leading to counter-intuitive behaviour when capturing changes in diversity. For example, an exact repetition of the reference data could be wrongly judged to be more diverse than a model that generates more samples with small but relevant deviations from the reference. Further, we argue that diversity is a multi-scale trend that should describe the effective size of a space across multiple levels of (dis)similarity rather than rely on fixed-scale snapshots. Indeed, summarising both the coarse and more granular geometry of the data manifold is necessary to get a complete picture of both local and global differences in entropy, clusterability and diversity.

This discussion thus points out a glaring need for more principled diversity measures. Addressing this, *magnitude functions* are particularly promising candidates for improved diversity measures that inherently satisfy all desiderata listed above, as shown in Appendix C.3. Many alternative summaries trivially fulfil a number of basic properties of diversity. However, it is non-trivial to satisfy *all* the desired axioms, making magnitude functions unique in their formulation. This axiomatic justification as well as our multi-resolution approach to diversity are the distinguishing characteristics and main advantage of our proposed diversity evaluation methods.

## 3.2 The Magnitude of a Metric Space

*Magnitude* is an invariant that measures diversity by describing the 'effective number of points' of a metric space as a function of its scaled distances [21].

**Definition 3.1** (Magnitude of a metric space). Let $X = \{x_1, \ldots, x_n\} \subseteq \mathbb{R}^D$ be a finite metric space with an associated distance metric $d$. For $1 \leq i, j \leq n$, the *similarity matrix* of $X$ is calculated as

$\zeta_X(i, j) := \exp(-d(x_i, x_j))$. If $\zeta_X$ is invertible, the *magnitude* of $X$ is defined as

$$\text{Mag}(X) := \sum_{ij}(\zeta_X^{-1})_{ij}. \tag{1}$$

The existence of magnitude is thus contingent on the existence of $\zeta_X^{-1}$. For negative definite metrics $d$ like the $L^1$ and $L^2$ distance, $\zeta_X$ is invertible [12]. Subsequently, we assume that $(X, d)$ permits the calculation of magnitude; in particular, $X$ must *not* have any duplicate points. While the magnitude of a metric space is expressive even at a *single* scale [5, 21, 23], magnitude unleashes its full potential in a *multi-scale* setting, assigning to a metric space not just a scalar but a function. To this end, we scale the distances in $X$, effectively viewing the same space through different lenses, or at different 'zoom factors,' for example by converting distances from centimetres to metres. Computing the magnitude for all such scales then yields the *magnitude function*.

**Definition 3.2** (Magnitude function). Let $(X, d)$ be a metric space and $tX := (X, d_t)$ its scaled version with $d_t(x, y) := t \cdot d(x, y)$ for a scaling factor $t \in \mathbb{R}_+$. The *magnitude function* of $(X, d)$ is the function $\text{Mag}_X : t \mapsto \text{Mag}(tX)$.

For $t \in (0, \infty)$, the magnitude function is defined for all but finitely many values of $t$ [21]. The magnitude function is also *continuous* [26, Corollary 5.5] for negative definite metrics.[1] For finite metric spaces, we have $\lim_{t \to \infty} \text{Mag}(tX) = |X| = n$, i.e. the *cardinality* of $X$ [21, Proposition 2.2.6]. This limit behaviour exemplifies to what extent the magnitude function describes the diversity of a space as 'the effective number of points at scale $t$.' Here, we extend magnitude functions to the domain $[0, \infty)$ by defining $\text{Mag}_X(0) := 1$.[2] Intuitively, this extension means that any metric space, when viewed from far away, looks like a single point. Notice that neither Definition 3.1 nor Definition 3.2 explicitly require specific properties of a metric (like the triangle inequality) and we find magnitude computable for generalised distance functions, including cosine distances, provided the similarity matrix $\zeta_X$ is invertible. Figure 1 illustrates how magnitude functions measure the effective number of distinct points for toy data, thus describing their diversity. Moreover, it provides an overview of our diversity evaluation framework, which we will now introduce.

## 3.3 Magnitude for Evaluating Diversity

As a multi-scale geometric invariant, magnitude can be extended to evaluate the diversity of latent representations. Here, we are studying a set of latent representations $\mathcal{X} = \{X_1, X_2, \dots\}$, where each $X_i \in \mathcal{X}$ is a finite subset of some latent space sharing the same notion of distance, e.g. $X_i \subseteq \mathbb{R}^D$. Given a latent representation $X \in \mathcal{X}$, e.g. a text, image, or graph embedding, we can use the $L^1$ or $L^2$ distance as a metric or semi-metrics like the cosine distance. Based on the choice of metric, we can interpret $\text{Mag}_X(t)$ as the effective number of points at scale $t$. In practice, this summarises how diverse points in the space are when observed at said scale factor. This multi-scale behaviour motivates us to propose a simple but expressive summary of a representation's magnitude function.

**Definition 3.3** (Area under the magnitude function, MAGAREA). Let $X$ be a metric space whose magnitude function $\text{Mag}_X(t)$ has been evaluated across the interval $T = [t_0, t_{\text{cut}}]$. We define the area under the magnitude function to be $\text{MAGAREA} := \int_{t_0}^{t_{\text{cut}}} \text{Mag}_X(t)dt$.

Moreover, we extend this proposed summary to measure the difference in diversity between two representations generated by the *same* (embedding) model. Notice that distances in these spaces are directly comparable and the respective magnitude functions can be compared across the same domain.

**Definition 3.4** (Magnitude function difference, MAGDIFF). Let $X$ and $Y$ be two metric spaces that share the same notion of distance. Assume the associated magnitude functions $\text{Mag}_X(t)$ and $\text{Mag}_Y(t)$ have been evaluated across the same interval $T = [t_0, t_{\text{cut}}]$. We define the magnitude function difference to be $\text{MAGDIFF} := \int_{t_0}^{t_{\text{cut}}} (\text{Mag}_X(t) - \text{Mag}_Y(t)) \, dt$.

Definition 3.3 and Definition 3.4 constitute novel multi-scale approaches for summarising and comparing magnitude functions, leading to theoretically well-founded diversity measures. MAGAREA

---

[1]$\text{Mag}_X$ is continuous for $t > t_{\text{crit}}$, where $t_{\text{crit}}$ is the supremum of its finitely many singularities.

[2]This assumes the so-called *one-point property*, i.e. $\lim_{t \to \infty} \text{Mag}_X(0) = 1$, which was shown to hold generically for almost all finite metric spaces [31].

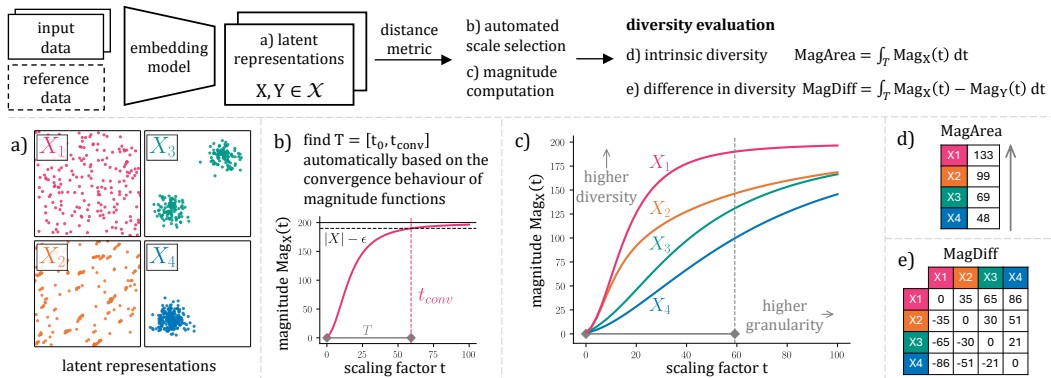

Figure 1: **Overview of our diversity evaluation pipeline.** (a) We start with an example of four latent spaces with 200 points, varying in diversity. (b) The magnitude function measures the effective number of points at $t$, a scale of distance between observations. When the scale factor $t$ almost equals zero, magnitude is close to 1, and a space effectively looks like one point. For large $t$, the number of effective points is noticeably higher and magnitude converges towards the cardinality. We find the approximate convergence scale, $t_{\mathrm{conv}}$, at which magnitude almost equals the cardinality, and use it to define the evaluation interval $T$ across which diversity changes most notably. (c) The more diverse the space, the higher the value of its magnitude function. By construction, $X_1$ is more diverse than $X_2$, $X_3$, and $X_4$, respectively, as we can see from the effective size of each space. We leverage this behaviour to define novel multi-scale indicators of diversity. (d) Our proposed measure of intrinsic diversity, MAGAREA, summarises the area under each magnitude function for *reference-free* diversity evaluation. (e) In a *reference-based* setting, we assess the difference in diversity using MAGDIFF, the area between two magnitude functions.

measures the cumulative value of magnitude summarising a space's intrinsic diversity while MAGDIFF measures the accumulated difference in diversity between two spaces. As we will later demonstrate in our experiments, integrating the changes in magnitude across a *range* of scale factors retains the desirable properties of single-scale magnitude, but yields more robust multi-scale summaries of diversity (see Appendix A.2 for an investigation of stability to perturbations). Furthermore, this comparison in terms of the effective number of points across scales remains directly interpretable.

### 3.4 Practical Usage and Implementation

In order to use our magnitude metric for reference-free and reference-based diversity evaluation, we obviate the choice of evaluation interval using knowledge about the convergence behaviour of magnitude functions. As a consequence, our magnitude-based diversity measures do not require manual parameter selection. First, we define a magnitude function's convergence scale.

**Definition 3.5** (Convergence scale, $t_{\mathrm{conv}}$)**.** Given a magnitude function $\mathrm{Mag}_X(t)$, we define its approximate convergence scale as $t_{\mathrm{conv}} \in \mathbb{R}$, with $\mathrm{Mag}_X(t_{\mathrm{conv}}) = |X| - \epsilon$ for some small $\epsilon > 0$. We set $\epsilon \leq 0.05|X|$ in this work.

This convergence scale thus indicates the resolution at which at least $95\%$ of observations are recognised by magnitude as being distinct. After reaching this convergence scale, we know that magnitude functions and hence diversity can increase by at most $\epsilon$ based on the convergence of magnitude towards the cardinality as illustrated in Figure 1. In practice, however, we find that *all* relevant changes in diversity happen at smaller scales of distance when individual points are not yet clearly separated. We thus choose the convergence scale defined in Definition 3.5 to be the upper bound of the evaluation interval $T$ to determine the most informative range of scales. We then find the convergence scale using numeric root-finding procedures as illustrated in Appendix B.2. When comparing the intrinsic diversity of multiple embeddings *without* a reference dataset, we compute MAGAREA across $T = [0, t_{\mathrm{cut}}]$ and choose $t_{\mathrm{cut}}$ to equal the median of the convergence scales of the embeddings. Taking the median here provides a stable compromise between the convergence behaviour of all functions. For *reference-based comparisons*, we simply calculate MAGDIFF, the difference between the magnitude functions, across $T = [0, t_{\mathrm{ref}}]$ where $t_{\mathrm{ref}}$ is the convergence scale of the reference embedding. In practice, we *approximate* the integrals in Definition 3.3 and

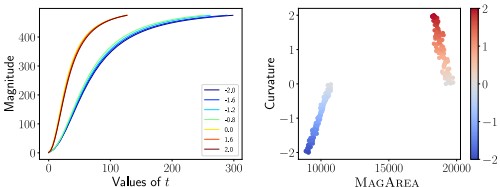

Figure 2: **Magnitude detects curvature.** Left: Magnitude functions for unit disks with varying curvature between $[-2, 2]$. Right: MAGAREA exhibits a linear relationship with curvature, indicating that it serves as a expressive predictor.

Table 1: **Magnitude estimates curvature.** MAGAREA outperforms more complex methods [41] using a *single feature*.

| Method | MSE ($\downarrow$) |
|---|---|
| SVR (selected PH features) | $0.27 \pm 0.07$ |
| SVR (PH vectorisation) | $0.17 \pm 0.05$ |
| SVR (all PH features) | $0.16 \pm 0.03$ |
| SVR (distance matrices) | $0.24 \pm 0.04$ |
| MLP (shallow) | $1.15 \pm 0.52$ |
| MLP (deep) | $1.56 \pm 0.68$ |
| MAGAREA (quantile) | $\mathbf{0.10 \pm 0.05}$ |
| MAGAREA (piecewise linear) | $\mathbf{0.05 \pm 0.03}$ |

Definition 3.4 via numerical integration across evenly-spaced scales sampled from the evaluation interval $T$. Choosing the number of scales is a trade-off between *accuracy* and *computational performance* as computational costs increase linearly with the number of times magnitude is evaluated. In terms of implementations, we also improve the efficiency of magnitude computations using a Cholesky decomposition (see Appendix A.5 for more details). Together with our automated scale-selection procedure, we thus overcome the main algorithmic hurdles that hitherto prevented the wider use of magnitude functions. Finally, we implement our methods in a Python package.[3]

### 3.5 Limitations

MAGDIFF is a reference-free measure of intrinsic diversity, but does not measure *fidelity*. It should therefore not be interpreted in isolation, but jointly with coverage-based metrics, for instance. Moreover, while we improve the efficiency of magnitude computations (see Appendix A.5) compared to previous implementations [5], thus making magnitude calculations feasible for practical analyses, novel approximation methods would be required to enable scaling to hundreds of thousands of observations. Finally, we focus on evaluating representation-based diversity and show that, given a latent representation, magnitude yields a better notion of diversity than current embedding-based methods. We do not investigate whether embedding-based similarities are outperformed by alternative task- or domain-specific similarities. Instead, our evaluation relies on the utility of embedding models and assumes that latent spaces encode useful/realistic relationships between samples.

## 4 Experiments

Our experiments demonstrate how magnitude leads to a better understanding of representational diversity. We show the following results: (i) Magnitude functions capture the curvature of a space. (ii) Magnitude functions are interpretable measures of the intrinsic diversity of embeddings, yielding superior results than other diversity measures when predicting the diversity of sentence embeddings across different text-generation tasks. (iii) Magnitude functions characterise and distinguish latent representations of large language models. (iv) Magnitude functions successfully detect mode dropping in distributions of image, and graph embeddings, while also reliably detecting mode collapse in graph embeddings. We subsequently use MAGAREA in reference-free settings to characterise intrinsic diversity (i, ii), while using MAGDIFF for reference-based comparisons (iii, iv).

### 4.1 Magnitude Functions Summarise Geometry

Magnitude functions encode the 'shape,' i.e. the geometry that is characteristic of the intrinsic data manifold, by capturing curvature and diversity. Curvature estimation is an important task in numerous domains like computer vision, computational geometry, and computer-aided design. The notion of curvature is inherently linked to diversity: The more positively curved a space is, the lower its diversity as points on the more curved surface move closer and closer together, thus decreasing its diversity. For specific examples of manifolds, magnitude can be expressed in terms of volume and total scalar curvature [44], a theoretical connection that we are the first to investigate empirically for

---

[3]The code for computing magnitude is available at `https://github.com/aidos-lab/magnipy`.

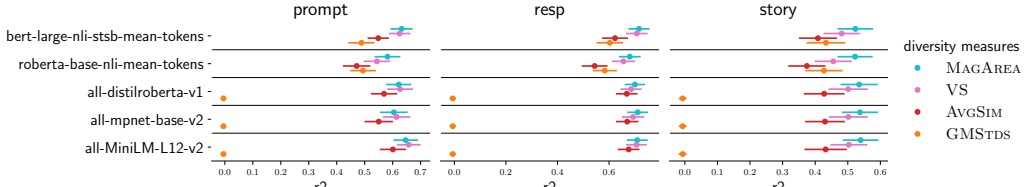

Figure 3: **MAGAREA outperforms alternative diversity measures** at predicting the ground truth-diversity of generated sentences, controlled by the softmax-temperature across 3 tasks and 5 embedding models. Baseline measures, AVGSIM and GMSTDS, perform worse in terms of the $R^2$ scores. Points show the mean of the $R^2$ scores, while lines represent the standard deviations across 5-fold cross-validation (repeated 10 times).

a broader class of spaces. Previous works have shown that alternative multi-scale methods, such as *persistent homology*, are able to detect curvature [4, 41]. Here, we demonstrate that the magnitude function is capable of achieving comparable performance, using simpler methods and only a single feature, namely MAGAREA. To this end, we generate a balanced dataset of point clouds of different curvature (following Turkes et al. [41] and detailed in Appendix D.1). We first assess to what extent the magnitude function can detect whether a unit disk has positive or negative curvature. Our main observation from plotting the functions for both groups in Figure 2 is that there is a clear separation between spaces of negative and positive curvature. We further test if we can predict curvature as a regression task. To this end, we try both piecewise linear and quantile regression,[4] using the area under the magnitude curve, MAGAREA, as a single feature. With 5-fold cross validation, we achieve an MSE of $0.05 \pm 0.03$ with the piecewise linear model and $0.10 \pm 0.05$ using quantile regression. Both scores substantially improve on previous methods [41] that made use of highly-sophisticated topology-based features and more heavily-parametrised deep learning models (see Table 1). These results underscore the expressivity and power of magnitude-based metrics, which enable us to solve the *same* task with a highly-simplified model. Moreover, this also demonstrates how magnitude describes the data manifold across multiple resolutions, motivating the use of magnitude functions as flexible, geometry-aware descriptors of diversity.

## 4.2 Magnitude Measures the Intrinsic Diversity of Text Embeddings

Next, we demonstrate the utility of using magnitude for intrinsic diversity evaluation and study its correspondence to known ground-truth diversity of text data. We analyse data from Tevet and Berant [38], consisting of 1K sets of 10 sentences each, generated for unique input prompts for 3 different sentence generation tasks, namely story completion (`story`), dialogue response generation (`resp`), and 3-word prompt completion (`prompt`). Per task, 10 response sets have been generated using the same decoding parameter, the softmax-temperature dec, which controls the diversity and randomness

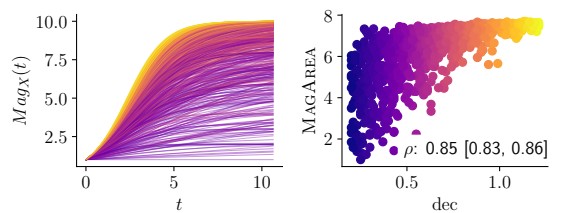

Figure 4: **MAGAREA correlates well with** dec **indicating the true diversity.** Here, we use `mpnet` embeddings for the `resp` dataset. $\rho$ denotes the rank correlation between MAGAREA and dec (95% bootstrap interval, 1000 resamples).

of the generated text. As dec decreases, models are skewed towards avoiding low-probability tokens. This leads to potentially higher quality and fidelity but lower diversity and creativity in generated text. We embed each set of responses using 5 pre-trained sentence transformer models [30], i.e. (1) `bert-large-nli-stsb-mean-tokens`, (2) `roberta-base-nli-mean-tokens`, (3) `all-mpnet-base-v2`, (4) `all-distilroberta-v1`, and (5) `all-MiniLM-L12-v2`. For each dataset and model, we compute the area under the magnitude function MAGAREA, evaluated until the

---

[4]Both models were chosen after explanatory analysis to offer multiple proposals on how to interpolate between the MAGAREA scores for surfaces of negative and positive curvature. The piecewise linear model better fits the trend in Figure 2, which is why it outperforms the quadratic relationship modelled via quantile regression.

median convergence scale across all embeddings as detailed in Section 3.4 using cosine distances. We compare this to the Vendi Score (VS), AVGSIM, and GMSTDS, calculated using cosine similarities. Moreover, we analyse the performance of each diversity metric at predicting the ground-truth diversity scores, dec, using 5-fold cross-validation repeated 20 times, trained via isotonic regression models;[5] and report their performance in terms of the coefficient of determination, $R^2$. Figure 4 depicts the positive rank correlation between magnitude and the softmax-temperature for one example setting, while Figure 3 shows results concerning the predictive performance of different diversity measures.

We observe that MAGAREA consistently outperforms alternative diversity measures computed from the same representations. MAGAREA achieves a median rank of 1 across experiments in terms of $R^2$ scores, followed by VS, AVGSIM and GMSTDS. Indeed, MAGAREA is most frequently the best-performing diversity measure for 77% of resamples when predicting decoding parameters, ranking second in the remaining cases. Meanwhile, VS most often achieves second place. This demonstrates the strength of MAGAREA as a theoretically-motivated and entropy-based measure of intrinsic diversity. By contrast, the baseline measure GMSTDS fails for any embedding that has at least one constant dimension, even reaching negative $R^2$ values for three of the five embedding models. This is followed by AVGSIM, which, while being less fallible than GMSTDS, simply measures average similarity and even ranks last across 27% of resamples. A further comparison of performance scores shows that MAGAREA outperforms AVGSIM by 0.12 higher mean $R^2$ scores on `story` and 0.07 on `resp` or `prompt` across embedding models. We find no dataset for which either AVGSIM or GMSTDS can be considered preferable predictors of the ground-truth diversity of text. Our results thus show the benefits of replacing simple summaries as the current standard for automated diversity evaluation with more sophisticated diversity measures like MAGAREA.

### 4.3 Magnitude Distinguishes and Characterises Embedding Models

Motivated by the capability of magnitude functions to encode representations, we now check whether the embedding spaces of different large language models can be distinguished via their intrinsic structure. To this end, we analyse 16384 documents of four different HuggingFace datasets, as embedded by Wayland et al. [43] using six different models (see Appendix D.3 for more details). We then either use PCA and normalisation to reduce each embedding space to 384 dimensions (to obtain a comparable dimensionality) or use the original embeddings without preprocessing. Further we subsample 300 documents at random from each space, repeating this procedure 200 times. Finally we use a 5-NN classifier to predict the embedding model based on the values of each diversity measure. This task is chosen to assess whether a simple classifier can distinguish embedding spaces solely based on their intrinsic diversity estimates. Table 2 reports the results of 5-fold cross-validation with 20 repetitions for both prepossessing choices. We either use Euclidean distances between single number summaries or, in the case of magnitude, use MAGDIFF directly as precomputed input distances for $k$-NN classification. We first observe that MAGDIFF best predicts the embedding model (with accuracies typically above 90%). Supplementary results in Table S.5 verify that these performance scores are almost identical for varying hyperparameter choices of $k$ neighbours. Surprisingly, the results further remain consistent for both pre-processing choices. This indicates that there are inherent differences in the structure and diversity of embedding spaces, which are preserved throughout dimensionality reduction and captured by magnitude. By using the difference between magnitude functions as a holistic summary, we once again surpass other summary statistics (which we observe to fail in distinguishing the smaller embedding models). Our results thus demonstrate that using MAGDIFF for comparing latent spaces across multiple scales is considerably more expressive than using single-number summaries of diversity.

### 4.4 Magnitude Evaluates Image Embeddings

*Mode dropping* is a common issue in generative modelling, referring to the inability of a model to capture all parts of an input distribution (for instance, a model trained to generate images of animals suffers from mode dropping if it can only generate images of dogs). To simulate this, we randomly sample 100 images from each of the 10 classes in CIFAR10 and embed them using a pre-trained Inception V3 model [37]. Subsequently, we re-sample increasingly more observations from *one* preferred image class. We either drop modes sequentially, or we move the same number of observations simultaneously from all other classes. Thus, diversity decreases gradually with the same

---

[5]We use these models to capture the non-linear monotonic relationship between dec and diversity.

Table 2: **Magnitude characterises text embedding models.** We show the accuracy (↑) of different diversity scores for distinguishing between six embedding models, using a 5-NN classifier.

| | | No pre-processing | | | PCA pre-processing | | | |
|---|---|---|---|---|---|---|---|---|
| Method
Dataset | MAGDIFF | AVGSIM | VS | GMSTDS | MAGDIFF | AVGSIM | VS | GMSTDS |
| cnn | **0.94 ± 0.02** | 0.87 ± 0.01 | 0.63 ± 0.01 | 0.66 ± 0.02 | **0.90 ± 0.02** | 0.88 ± 0.02 | 0.67 ± 0.03 | 0.66 ± 0.03 |
| patents | **0.99 ± 0.01** | 0.92 ± 0.01 | 0.63 ± 0.02 | 0.66 ± 0.02 | **0.96 ± 0.01** | 0.91 ± 0.02 | 0.64 ± 0.03 | 0.66 ± 0.03 |
| arXiv | **0.99 ± 0.01** | 0.89 ± 0.01 | 0.78 ± 0.01 | 0.66 ± 0.02 | **0.99 ± 0.01** | 0.88 ± 0.02 | 0.78 ± 0.02 | 0.66 ± 0.03 |
| bbc | **0.98 ± 0.01** | 0.74 ± 0.01 | 0.84 ± 0.02 | 0.66 ± 0.02 | **0.95 ± 0.01** | 0.73 ± 0.03 | 0.84 ± 0.02 | 0.66 ± 0.03 |

'speed' across both procedures, but fidelity should not change. We treat each class as the preferred image class twice, leading to 20 re-samples per mode dropping scenario [28]. Our analysis compares the changes in recall and coverage, setting the number of nearest neighbours to $k = 10$. Further, we calculate the relative change in $\mathrm{Mag}(0.5t_{\mathrm{ref}})$, i.e. magnitude computed at half the convergence scale of the reference using Euclidean distances. Similarly, MAGDIFF is the difference between the magnitude functions relative to the area under the reference magnitude function.

Figure 5 shows the changes in diversity as modes are being dropped. Ideally, every diversity measure should show the *same* decrease in diversity, irrespective of resampling strategy. However, we observe that both recall and coverage wrongly assess that diversity decreases faster during sequential resampling. Even worse, coverage only detects simultaneous mode dropping after around 70% of all points have shifted to one mode. This undesirable behaviour of both metrics is caused by their reliance on a fixed neighbourhood size for approximating the underlying manifold, thus overestimating the extent to which the perturbed samples

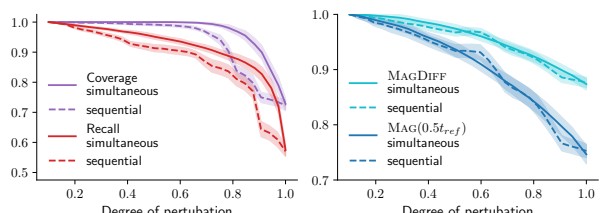

Figure 5: **Magnitude correctly detects that diversity decreases in the same manner across simultaneous and sequential mode dropping** outperforming recall and coverage. Lines show the mean values of each metric across 20 resamples, shaded areas the standard deviations.

reflect the diversity of the reference distribution. In comparison, MAGDIFF as well as magnitude evaluated at a single scale both successfully measure the gradual decrease in diversity across both mode dropping scenarios.

### 4.5 Magnitude Evaluates Graph Generative Models

Diversity evaluation in graph learning is fraught with difficulties, in particular when aiming to detect common problems like *mode collapse* or *mode dropping* [29, 40]. In the following, we will study graph generative models (GGMs), which take a set of input graphs and generate new samples that should follow the *same* distribution. The question that we aim to answer here is whether our proposed magnitude-based metric is more expressive in capturing the diversity of the generated graphs than classical metrics like *maximum mean discrepancy* (MMD) and measures inspired from evaluating image generative models (precision, recall, coverage, density). To this end, we analyse 3 synthetic (Lobster, Grid, and Community) and 2 real-world (Proteins and Ego) graph datasets, and compute commonly-used evaluation metrics [29, 40] as detailed in Appendix D.5. To test the diversity of generated samples, we replicate the experimental setup of Thompson et al. [40] and add our own measure, MAGDIFF computed using $L^2$ distances from Graph Isomorphism Network [45, GIN] embeddings with varying hyperparameters. For the *mode collapse* experiments, we substitute each embedded graph with its cluster centre. Thus, the degree of perturbation $p$ equals the proportion of clusters collapsed in this manner. The larger the value of $p$, the more clusters have been perturbed decreasing the diversity. For the *mode dropping* experiments, we remove clusters, and keep the size of the generated dataset the same as the reference by randomly resampling from the remaining classes.

Figure 6 shows the results of both *mode collapse* and *mode dropping* for the Lobster dataset. We observe similar trends across all datasets, but have chosen this dataset as a running example. Ideal measures should exhibit high rank correlation to the degree of perturbation, indicating that they

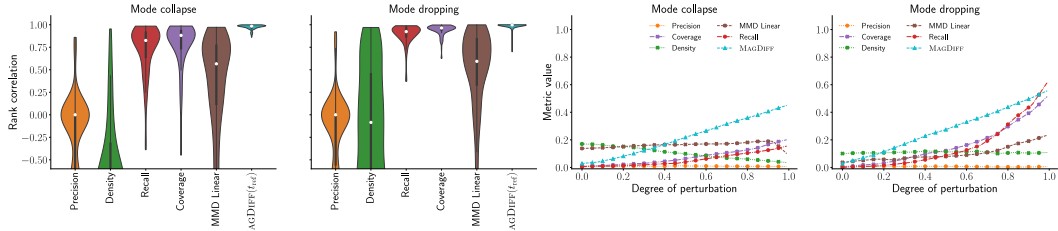

Figure 6: **MAGDIFF outperforms existing graph diversity metrics at detecting mode collapse and mode dropping.** We report the Spearman correlation between each metric and the degree of perturbation $p$ for the Lobster dataset (the same pattern holds for Proteins, Community, Ego, Grid, see Appendix D.5). Violin and box plots show the distributions across different hyperparameter choices. Measures that capture the decrease in diversity accurately should increase as a function of $p$. Rank correlation of 1 corresponds to an ideal metric. Our metric best captures the changes in diversity for both mode dropping and collapse.

are capable of capturing the decrease in diversity properly, i.e. as a function of $p$. We note that in contrast to our magnitude-based metric, *recall* and *coverage* exhibit worse results, as evidenced by their lower mean correlation coefficient. Despite being specifically designed to measure the diversity of a dataset [40], they only catch up to our magnitude metric when the degree of perturbation $p$ is around 0.9 (see Figure 6, right-hand plots). Magnitude dominates in the majority of the values of $p$ best showing the steady decrease in diversity, while recall and coverage become more sensitive for exceedingly large values of $p$, i.e. in unrealistic situations where most of the modes have been dropped. Moreover, their performance is highly contingent on $k$, the parameter used to construct a $k$-NN graph for computing these neighbourhood-based metrics. Magnitude functions meanwhile give more holistic summaries of both local and global patterns in diversity. Please refer to Figure S.16 for the aggregated results over all datasets, which exhibit a similar pattern (in that our metric outperforms both *recall* and *coverage*).

## 5 Discussion

We have proposed novel diversity measures for evaluating latent representations. Our measures are based on *metric space magnitude*, a multi-scale invariant summarising geometrical characteristics of the input data. We have demonstrated axiomatically and empirically that our magnitude-based measures are superior to current baseline measures of intrinsic diversity. In a reference-free scenario, we observe that magnitude outperforms alternative measures when predicting the ground truth diversity for text embeddings. Given a reference dataset, we find that magnitude captures mode collapse and mode dropping better than existing metrics for evaluating generative models for both image and graph modalities. Furthermore, we have shown that magnitude can measure the intrinsic curvature of input data, outperforming previous methods. Magnitude thus gives a provably stable, unsupervised diversity metric that can be computed efficiently and allows users to flexibly choose a notion of dissimilarity. For future work, we believe that magnitude exhibits a strong potential for applications to unaligned spaces with varying notions of distances. Moreover, we believe that integrating magnitude into deep learning models would be beneficial for obtaining novel diversity- and geometry-based regularisation strategies.

## Acknowledgements

The authors are grateful for the stimulating discussions with the anonymous reviewers and the area chair, who believed in the merits of this work. We further want to thank Dr. rer. nat. Dr. iur. Corinna Coupette, Emily Simons, and Jeremy Wayland for their help in proofreading the paper. K.L. is supported by the Helmholtz Association under the joint research school 'Munich School for Data Science (MUDS).'

## Funding Disclosure

K.L. gratefully acknowledges support from the 2024 NeurIPS Financial Assistance Program. R.A. was supported by (i) the United Kingdom Research and Innovation (grant EP/S02431X/1), UKRI Centre for Doctoral Training in Biomedical AI at the University of Edinburgh, School of Informatics, (ii) the International Helmholtz–Edinburgh Research School for Epigenetics (EpiCrossBorders), and (iii) a Helmholtz Visiting Researcher Grant. B.R. was partially supported by the Bavarian state government with funds from the *Hightech Agenda Bavaria*. This work has received funding from the Swiss State Secretariat for Education, Research, and Innovation (SERI). The authors declare no competing interests. The funders had no role in the preparation of the manuscript or the decision to publish.

## Impact Statement

This paper presents work whose goal is to advance the evaluation diversity in representation learning, leading to increased fairness and trustworthiness in model evaluation. While representational diversity in terms of model outputs may have potential negative impacts, depending on the task at hand, we feel there are none that need to be specifically highlighted here. However, we acknowledge the potential for societal harm if our notion of representational diversity is confused with the meaning of diversity in the colloquial or societal context, which is admittedly even harder to measure and requires a larger discussion involving all affected communities.

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

# Appendix (Supplementary Materials)

# A   Extended Theory and Empirical Validation

## A.1   Illustration of Magnitude and Magnitude Weights

While we did not use magnitude weights, which are the individual contribution of each point in a space to its overall magnitude, throughout our experiments, they play a more central role in some of the later proofs and the computation of magnitude in practice. Further, magnitude weights give an intuitive explanation on how each individual observation influences magnitude and the magnitude function as illustrated in Figure S.1.

**Definition A.1** (Magnitude weights)**.** Let $X = \{x_1, \ldots, x_n\}$ be a finite metric space with an associated distance metric $d$. The *similarity matrix* of $X$ is defined as $\zeta_X(i, j) = \exp(-d(x_i, x_j))$ for $1 \leq i, j \leq n$. If $\zeta_X$ is invertible, the *magnitude weighting vector* $w_X$ is defined as $w_X := \zeta_X^{-1} \mathbb{1} = \mathbb{1}^\top \zeta_X^{-1}$. Denoting the $i$th element of $w_X$ by $w_{x_i}$, we obtain an equivalent characterisation of magnitude as $\mathrm{Mag}(X) = \sum_i w_{x_i}$

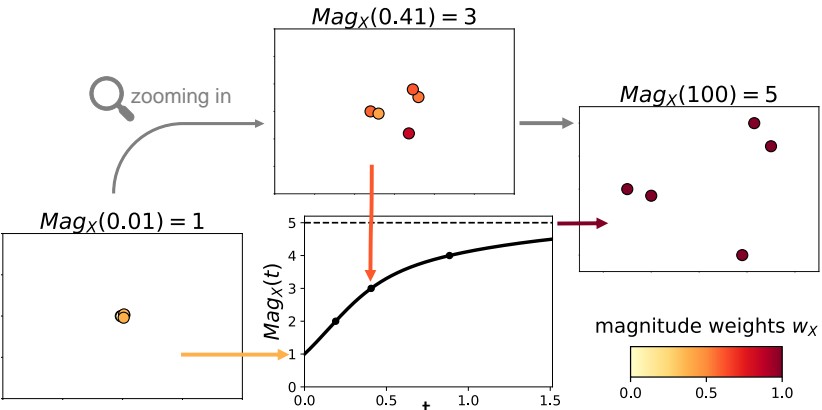

Figure S.1: **Example of magnitude weights and the magnitude function for a metric space with 5 points.** When the scaling factor $t$ is very small, e.g. $t = 0.01$, the magnitude weights of all points sum up to approximately 1, so that magnitude is very close to 1, and the space effectively looks like one point. Following this, as we zoom in further, magnitude grows and at $t = 0.41$, 3 distinct clusters or points are visible. Finally, for $t = 100$, all the points are clearly separated, their magnitude weights converge to one, and the value of magnitude approaches 5, i.e. the cardinality of the space.

## A.2   Stability Proof

Next to the theoretical properties linking magnitude to geometrical properties of a space, which we previously outlined, we further prove that magnitude, as a metric space invariant, also satisfies properties that are advantageous in the setting of analysing latent representations. Specifically, we prove that magnitude and thus the proposed magnitude differences satisfy certain *stability properties* in light of perturbations of metric space. By this, we mean that if two metric spaces $X, Y$ are *close*, we want to obtain bounds on the differences between their magnitude values. The canonical choice to measure closeness would be the Gromov–Hausdorff distance, but in the absence of strong results concerning the behaviour of magnitude under this distance [21], we resort to a more general—but also weaker—notion of similarity in terms of *continuity*. More precisely, we will show that the similarity matrices used in the calculation of magnitude are well-behaved in the sense that closeness of metric spaces (under some matrix norm) translates to a continuous bound on the variation of the similarity matrices. We first prove a general result about matrices and their associated transformations.

**Lemma A.2.** *Let* $\|A\|_2 := \sup \{\|Ax\|_2 : x \in \mathbb{R}^n \text{ with } \|x\|_2 = 1\}$ *refer to the* induced 2-norm *for matrices, and let $A, B$ be two $n \times n$ matrices with $\|A - B\|_2 \leq \epsilon$. Moreover, let $f(M) := \mathbb{1}^\top M \mathbb{1}$. Then $\|f(A) - f(B)\|_2 \leq n\epsilon$.*

*Proof.* Because $\|\cdot\|_2$ is a *consistent* norm, we have $\|f(M)\|_2 \leq \|\mathbb{1}^\top\|_2 \|M\|_2 \|\mathbb{1}\|_2 = n\|M\|_2$ for all $n \times n$ matrices $M$. Without loss of generality, assume that $\|f(A)\|_2 \geq \|f(B)\|_2$ and $\|A\|_2 \geq \|B\|_2$. Thus, $\|f(A)\|_2 - \|f(B)\|_2 \leq d(\|A\|_2 - \|B\|_2) \leq d(\|A - B\|_2) = n\epsilon$. $\qquad\square$

Treating $A, B$ as inverse similarity matrices, the preceding statement shows that if the two inverse similarity matrices are close with respect to their spectral radius, the difference between their magnitude can be bounded. The following lemma shows that the similarity matrices satisfy a general continuity condition.[6]

**Lemma A.3.** *Let $(X, d_X)$ and $(Y, d_Y)$ be two metric spaces with corresponding distance matrices $D_X, D_Y$ and cardinality $n$. For all $\epsilon > 0$, there exists $\delta > 0$ such that if $|D_X - D_Y| < \delta$ holds elementwise, then $\|\zeta_X - \zeta_Y\|_2 \le \epsilon$.*

*Proof.* As a consequence of the continuity of the exponential function, we know that there is $\delta$ such that $|\zeta_X - \zeta_Y| < n^{-1}\epsilon$. The row sums of $\zeta_X - \zeta_Y$ are therefore upper-bounded by $\epsilon$. We thus have $\|\zeta_X - \zeta_Y\|_2 \le \epsilon$ [27, Theorem 1.1, p. 24]. $\square$

As a consequence of Lemma A.3, and the continuity of matrix inversion, we know that magnitude is well-behaved under small perturbations of the respective distance matrices. Given a pre-defined threshold $\epsilon$, we can always find perturbations that preserve the magnitude difference accordingly. Notice that this result does not make any assumptions about the Gromov–Hausdorff distance of the metric space and only leverages the distance matrices themselves. Moreover, this result applies in case $X, Y$ are close with respect to the *Hausdorff distance*. If $d_H(X, Y) < \delta$, the elementwise condition $|D_X - D_Y| < \delta$ is satisfied *a fortiori*. This stability of single-scale magnitude then further ensures the stability of the difference between magnitude functions as defined in 3.4 in the same sense. Nevertheless, from a theoretical point of view, this result could be made stronger by showing bounds in terms of distances between the metric spaces. We leave such a result for future work, noting in passing that such strong results remain elusive at the moment [14]; it is known, however, that the magnitude function is at least *lower semicontinous* [25, Theorem 2.6].

## A.3  Empirical Stability

We further investigate the empirical stability of the magnitude function difference. Given the difficulty in proving strong theoretical stability results, we verify that, in practice, the magnitude function difference remains stable when adding noise to the input space. We thus sample points from a Laplace distribution with mean $\mu = 0$ and variance $2b^2$ with different levels of noise, i.e. $b \in \{0.0001, 0.001, 0.005, 0.01, 0.05\}$. Figure S.2 depicts the errors in magnitude function difference relative the the area under the magnitude function of the unperturbed data across three different datasets (circles, Swiss Roll, Gaussian blobs), using a different number of samples (varying between 100 and 5000 across 50 repetitions). The bound of 5000 points has been chosen given the clear downwards trend across multiple noise levels; we expect the same trend to hold for larger sample sizes. We observe that the magnitude function difference does not increase above the value of $1 \times 10^{-3}$ with increasing sample size. In fact, the difference fluctuates more for smaller number of points, but this is still within a very small range. We therefore conclude that the magnitude function difference between the original space and its noisy version does not change much, which indicates that our measure is reliable and stable across multiple experimental conditions.

## A.4  Isometry Invariance

A measure of the difference (in diversity) between latent spaces should fulfill certain desirable properties from both practical and theoretical perspectives. In the following we will show a minimum requirement, namely that the magnitude difference between isomorphic spaces equals zero.

**Definition A.4** (Isometry). Let $(X, d_X)$ and $(Y, d_Y)$ be two metric spaces. A map $f \colon X \to Y$ is called an *isometry*, or distance-preserving, if for any $a, b \in X$, we have $d_X(a, b) = d_Y(f(a), f(b))$. $X$ and $Y$ are called *isometric* if there is a *bijective isometry* from $X$ to $Y$.

**Lemma A.5** (Isometry invariance). *Given two isometric spaces $X, Y$, we have $\mathrm{Mag}_X = \mathrm{Mag}_Y$.*

*Proof.* Let $(X, d_X)$ and $(Y, d_Y)$ be metric spaces with cardinality $n$ and let $f \colon X \to Y$ denote their isometry. Then, the similarity matrix of $X$ is $\zeta_X(i, j) = \exp(-d_X(x_i, x_j))$. Since $f$ is

---

[6]It is clear that the mapping itself is continuous because of the functions involved in its calculation. However, we find it important to remark on the bound obtained with respect to the *spectral norm* of the two similarity matrices.

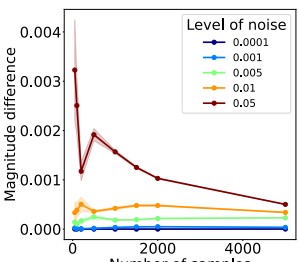 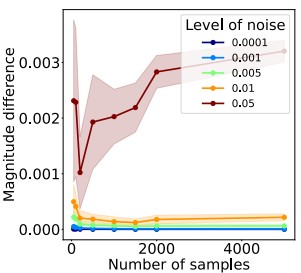 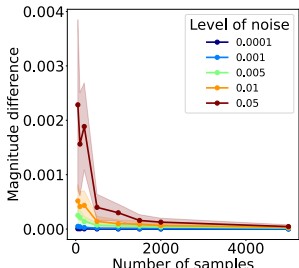

Figure S.2: **Empirical stability of magnitude.** Magnitude difference is stable across different datasets (from left to right: Circles, Swiss Roll, Gaussian blobs) and sample sizes. The lines show the mean magnitude difference relative to the magnitude area of the unperturbed data and the shaded area the standard deviation calculated across 50 repetitions.

an isometry, we have, $d_X(x_i, x_j) = d_Y(f(x_i), f(x_j))$. Hence, $\zeta_X(i,j) = \exp(-d_X(x_i, x_j)) = \exp(-d_Y(f(x_i), f(x_j))) = \zeta_Y(i,j)$. Since $X$ and $Y$ have the same similarity matrix, we have $\mathrm{Mag}_X = \mathrm{Mag}_Y$. □

**Corollary A.6.** *The magnitude functions of two isometric spaces $X, Y$ are equal for all $t \geq 0$.*

Notice that the *converse* of this statement is not true in general, i.e. there are non-isometric spaces whose magnitude functions are the same [21].

**Corollary A.7.** *Let $X$ be a metric space and $Y = cX$ with $c \in \mathbb{R}_+$. Then the magnitude functions of $X$ and $1/cY$ are equal. Also, the magnitude functions of $1/\mathrm{diam}_X X$ and $1/\mathrm{diam}_Y Y$ are equal, where $\mathrm{diam}_X := \max(d_X)$.*

**Corollary A.8.** *Magnitude function difference equals zero for isomorphic spaces.*

### A.5 Computing Magnitude

A naïve calculation of magnitude according to Definition 3.1 requires inverting the similarity matrix $\zeta_X$, which has a worst-case complexity of $\mathcal{O}(n^3)$ and is numerically unstable. However, inverting $\zeta_X$ is not required in practice; instead, it suffices to solve certain *linear equations* as also pointed out by Huntsman [17]. First, we notice that the calculation of magnitude can be written as $\mathrm{Mag}(X) := \mathbb{1}^\top \zeta_X^{-1} \mathbb{1}$. For finite metric spaces and negative definite metrics, $\zeta_X$ is a *symmetric positive definite matrix*, thus affording a *Cholesky decomposition*, which factorises $\zeta_X = LL^\top$, with $L$ being a *lower triangular matrix*. This operation is numerically stable and more efficient than matrix inversion [16]. We thus have $\mathrm{Mag}(X) := \mathbb{1}^\top \zeta_X^{-1} \mathbb{1} = \mathbb{1}^\top (LL^\top)^{-1} \mathbb{1} = (L^{-1}\mathbb{1})^\top (L^{-1}\mathbb{1})$. This is equivalent to calculating $x^\top x$ with $x = L^{-1}\mathbb{1}$, which we can efficiently obtain by solving $Lx = \mathbb{1}$ since $L$ is lower triangular. Likewise, we can reformulate the calculation of the *magnitude weight vector* $w_X = \zeta_X^{-1} \mathbb{1}$ as solving $\zeta_X w_X = \mathbb{1}$, which also benefits from the Cholesky factorisation.

### A.6 Benchmarking Computational Times

To assess the improvements in computational efficiency discussed in Appendix A.5, we benchmark the following computational methods in Python:

- Numpy inv: Inversion of the whole matrix $\zeta$ using `numpy.linalg.inv` as suggested by Bunch et al. [5]; see also https://github.com/AmFamMLTeam/metric-space-magnitude for an implementation.
- Scipy solve: Solving for the magnitude weights using `scipy.linalg.solve` and assuming $\zeta$ to be positive definite.
- Cholesky weights: Cholesky decomposition using `scipy.linalg.cho_factor` to compute the magnitude weights.
- Cholesky: Using a Cholesky decomposition as suggested above to compute the value of magnitude directly. This is the method we implemented to compute magnitude throughout this work.

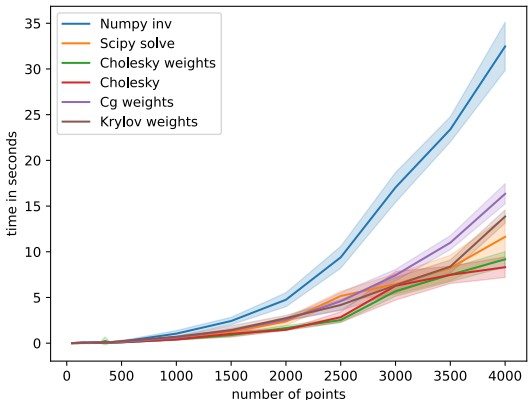

Figure S.3: **Benchmark of computational times in seconds for magnitude functions evaluated across** 10 **scales.** We observe that Cholesky decomposition performs well, even for larger number of observations ensuring that magnitude functions can be computed in a matter of seconds. Lines show the time in seconds across five repeats, shaded areas the standard deviations.

- Cg weights: Conjugate gradient iteration using `scipy.sparse.linalg.Cg` and an absolute tolerance of 1e-3 to solve for the magnitude weights.
- Krylov weights: Pre-conditioned conjugate gradient iteration using `krypy.linsys.Cg` as implemented by Salim [33] to calculate magnitude weights.

All methods are evaluated on simulated data of a Swiss Roll with an increasing number of points. For each space, magnitude is evaluated at ten scales evenly spaced between zero and the convergence point. The computational times and their standard deviations are recorded across five re-runs in Figure S.3. Results clearly show that naive inversion of the whole similarity matrix is by far the most costly method for computing magnitude. This is followed by the two conjugate gradient methods described above, where the pre-conditioned version is somewhat faster than the implementation without pre-conditioning for larger numbers of points. However, for evaluating magnitude at only 10 scales these approaches do not necessarily lead to improved performance compared to solving for the weights simply using `scipy.linalg.solve`. Finally, we note that our proposed implementation using Cholesky decomposition is the fastest computational method achieving less than a third of the computational time of the most naive implementation for larger datasets. Indeed, these results confirm that even for thousands of points magnitude functions are efficiently computable in a matter of seconds. Overall, this computational performance is more than sufficient for the relevant diversity evaluation tasks discussed in this study. State-of-the-art graph datasets are typically small, the output of text generation models is often assessed on specific tasks and even image embeddings are frequently evaluated in terms of meaningful subsets, e.g. by studying intra-class diversity. Indeed, we ran all our experiments locally with the following hardware specifications:

- **CPU:** 12th Gen Intel(R) Core(TM) i7-1265U,
- **RAM:** 16 GB DDR4,
- **SSD:** 512 GB NVMe SSD

## B  Additional Details for Our Methods

### B.1  Embedding Data

Creating latent representation or embedding $X = M(I)$, whose diversity should be evaluated, depends on the complexity of the specific model $M$ and the input data $I$ that should be represented. This step is independent of our design choices. Given a latent representation we then choose a suitable notion of distance, for example cosine distances are a natural choice for text embeddings, which (compared to e.g. Euclidean distances) better represent similarity in meaning rather than text lengths; or Euclidean distances to understand more general latent spaces.

## B.2 Scale-Finding Procedure

Next, we give a brief illustration of the scale-finding procedure developed to automate magnitude computations. We use TOMS 748 root-finding algorithm [2] as implemented via `scipy.optimize.toms748`. Our aim is to find the scale $t_{\text{conv}}$ at which $\text{Mag}_X(t_{\text{conv}}) \approx |X| - \epsilon$ such that $|t_{\text{true}} - t_{\text{conv}}| \leq \text{atol} + \text{rol} \cdot |t_{\text{conv}}|$ where $t_{\text{true}}$ is the true scale at which $\text{Mag}_X(t_{\text{conv}}) = |X| - \epsilon$, rtol is the relative error and atol the absolute error. We initialise the algorithm at the search interval $[a, b]$ setting $a = 0$ and $b = 100$ as an initial guess. We then check if $(\text{Mag}_X(a) - |X| + \epsilon)$ and $(\text{Mag}_X(b) - |X| + \epsilon)$ have opposite signs. If not we update $a = b$ and $b = 100 \cdot b$ repeating this at most 100 times and raising an error if they still share the same sign indicating the root-finding failed. Otherwise, we run TOMS 748 algorithm to find $t_{\text{conv}}$ as specified above using at most 100 iterations. This algorithm requires $\text{Mag}_X(t_{\text{conv}})$ to be continuous to perform reliably, which holds for negative definite metric spaces $X$ as proven in Appendix A.2.

After reaching the convergence scale as defined in Definition 3.5 we know that magnitude and hence diversity can change by at most $\epsilon$, which directly follows from the convergence behaviour of magnitude. Based on this, the default parameter $\epsilon = 0.05|X|$ or $\epsilon = 0.01|X|$ is chosen as a sensible compromise for determining the scales of interest across which diversity changes most notably. To support this choice of convergence scale, we empirically investigate the impact it has on the results reported in our work. In particular, Figure S.12 investigates how the choice of $\epsilon$ influences the correspondence between human evaluation scores and the diversity of generated text, while Figure S.17 outlines the results of the graph generative model evaluation experiment for varying values of $\epsilon$. We note that choices of $\epsilon \leq 0.05|X|$ give stable and generally very good results for both reference-free and reference-based diversity evaluation as further detailed in Appendix D.

Figure S.4 gives some further intuition on this scale finding procedure. In particular, it demonstrates how taking the median convergence scale across four example spaces gives a suitable evaluation interval across which their magnitude can be compared as explained in more detail in Section 3.4. This simple example also illustrates why to compare latent spaces in a reference-free setting we recommend using the median or another suitable quantile of the converge scales rather than the minimum or the maximum, which are less robust and more sensitive to outliers.

Finally, we note that the scale-finding approach can not only be used to compute MAGAREA or MAGDIFF, but also to find a suitable scale at which to evaluate magnitude at a single resolution, leading to a summary of diversity at a single threshold. In practice, we recommend choosing this single scale to be less or equal to the convergence scale of a space and that the best resolution to choose depends on the question of interest. Lower scales give a more coarser view summarising the diversity of large clusters while higher scale parameters show a clearer separation between individual observations.

## B.3 Integration

In practice, we evaluate the integral from Definition 3.4 and Definition 3.3 using Trapezoidal integration as implemented via `scipy.integrate.trapezoid` across a certain number, $n_{\text{ts}}$, of evenly spaced evaluation scales in $T = [t_0, t_{\text{cut}}]$. This numerical integration method is chosen due to its simplicity and computational efficiency, but more complex approximation methods can also be employed.

# C Extended Discussion on Diversity Measures

## C.1 Definitions of Intrinsic Diversity Measures

The difficulty in defining diversity in representation learning has led to a few varying proposals for evaluating the intrinsic diversity of latent representations. Amongst these we consider the following three methods as baseline measures:

**GMSTDS:** For a $X$, a D-dimensional embedding, it is directly computed as

$$\text{GMSTDS} = \sqrt[D]{\prod_{i=j}^{D} \sigma_j} \tag{2}$$

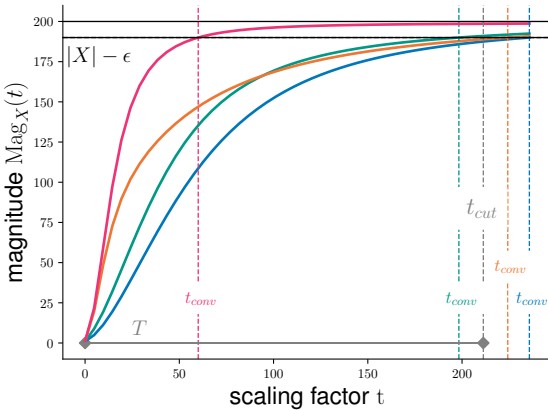

Figure S.4: **Illustration of the proposed scale finding procedure** for comparing the diversity of four examples from Figure 1. We compute the approximate convergence scale for each space as defined in Definition 3.5 at which magnitude has reached a certain value $|X| - \epsilon$ close to the cardinality. We then take the median of these scales to find a shared endpoint $t_{\text{cut}}$ for defining a shared evaluation interval $T = [0, t_{\text{cut}}]$.

where $\sigma_j = \sqrt{\frac{1}{n} (\sum_{i=1,..,n} x_{ij} - \hat{x}_j)^2}$ is the standard deviation across the j-th embedding dimension [20]. Thus, GMSTDS regards an embedding as a cluster and assessing diversity by quantifying its spread.

**AVGSIM:** Average mean similarity (or variations of it) is the most frequently used diversity measure in ML. It is simply computed as

$$\text{AVGSIM} = \frac{1}{\binom{n}{2}} \sum_{i,j \leq n, j > i} \zeta(i,j) \tag{3}$$

across all distinct pairs of points in $X$ assuming $\zeta$ is symmetric [38]. This approach simply summarises that in a more diverse space, observations should on average be less similar.

**Vendi Score (VS):** We also consider the Vendi Score, which is the only entropy-based diversity measure proposed in related ML literature. Let $\zeta$ be a positive semi-definite similarity matrix with $\zeta(i,i) = 1$ for all $i \leq n$. Compute $\lambda_i$, the eigenvalues of $\zeta/n$. Then the Vendi Score is defined as

$$\text{VS} = \exp(-\sum_{i=1}^{n} \lambda_i \log(\lambda_i)) \tag{4}$$

taking $0 \log(0) = 0$ by convention. That is, the Vendi Score is the exponential of the Shannon entropy of the eigenvalues of $\zeta/n$ [13]. It can thus be interpreted as summarising the effective number of modes in a space at a specific scale of similarity.

## C.2 Defining Reference-based Evaluation Metrics

Here we define the metrics used in the image and graph embedding experiments.

$$\text{Precision} = \frac{1}{M} \sum_{j=1}^{M} 1_{Y_j \in \text{manifold}(X_1,...,X_N)} \tag{5}$$

$$\text{Recall} = \frac{1}{M} \sum_{i=1}^{N} 1_{X_i \in \text{manifold}(Y_1,...,Y_M)}, \tag{6}$$

where the manifold is defined as $\text{manifold}(X_1, \ldots, X_N) = \bigcup_{i=1}^{N} B(X_i, \text{NND}_k(X_i))$. Here, $B(x, r)$ is the sphere in $\mathbb{R}^d$ of radius $r$ around $x$, $\text{NND}_k(X)$ is the distance to the $k$-th nearest neighbour, and $1_{(\cdot)}$ is the indicator function.

Similarly, density and coverage are defined as follows:

$$\text{Density} = \frac{1}{M} \sum_{j=1}^{M} \sum_{i=1}^{N} 1_{Y_j \in B(X_i, \text{NND}_k(X_i))}, \tag{7}$$

$$\text{Coverage} = \frac{1}{N} \sum_{i=1}^{N} 1_{\exists j \text{ s.t.} Y_j \in B(X_i, \text{NND}_k(X_i))}, \tag{8}$$

Maximum Mean Discrepancy (MMD) [15, 46] is a metric based on graph statistics. MMD Linear is computing MMD with a linear kernel [29]. We define $\mathbb{S}_r = \{x_1^r, \dots, x_m^r\} \sim P_r$ and $\mathbb{S}_g = \{x_1^g, \dots, x_n^g\} \sim P_g$, where $x_i$ is a feature vector from a corresponding graph $G_i$. Therefore, MMD is defined as follows:

$$\text{MMD}(\mathbb{S}_r, \mathbb{S}_g) = \frac{1}{m^2} \sum_{i,j=1}^{m} (k(x_i^r, x_j^r)) + \frac{1}{n^2} \sum_{i,j=1}^{n} (k(x_i^g, x_j^g)) - \frac{2}{nm} \sum_{i=1}^{n} \sum_{j=1}^{m} (k(x_i^g, x_j^r)), \tag{9}$$

where $k(\cdot, \cdot)$ is a general kernel function. For the case of the metric MMD Linear, used in our graph experiments, we use a linear kernel.

## C.3    An Axiomatic Approach to Defining Intrinsic Diversity

The attempt to define, prove and interpret a theoretically well-founded notion of diversity has inspired decades of heated debate in theoretical ecology. While in the context of machine learning, diversity is still very seldomly explored axiomatically, measures of biodiversity are often defined in a more well-established and unified framework built on mathematically complex ideas, such as entropy and extended notions of size. However, there is a lot of benefit to be gained by a more theoretical discussion on evaluating diversity in representation learning. Indeed, as pointed out by Leinster [22], if a diversity measure does not pass basic logical tests, it is likely to be misleading and potentially useless for practical applications, which can have far-reaching detrimental consequences. By discussing some of these fundamental properties we thus uncover exactly how existing diversity measures for evaluating latent representations fail at essential requirements. We further discuss how magnitude functions improve upon alternative diversity metrics.

Throughout this section let $m(X) \in \mathbb{R}$ be a diversity measure of the (metric) space $X$, let $d_X$ be a distance metric on this space and $\zeta_X$ a matrix of pairwise similarities.

**Diversity measures the absolute richness in observations.** Metric space magnitude as a diversity measure has been introduced with the following three properties fundamental properties in mind [36]:

- **Monotonicity in observations.** Including a new observation with all positive distances to a metric space with a negative definite metric does not decrease diversity. Formally, for all $x_0 \notin X$ where $(X, d_X)$ is a metric space define $(Z = X \cup x_0, d_Z)$ via inclusion so that $d_Z$ is a valid metric. If $d_Z(x_0, x) > 0$ for all $x \in X$, the diversity measure $m$ is monotone in observations if $m_X \leq m_Z$. For magnitude, when taking $m = \text{Mag}$, this directly follows from Corollary 2.4. in Leinster [21].
- **Twin property.** Diversity does not change when including a duplicate observation. Formally, for $x_0 \in X$ define $Z = x_0 \cup X$. A diversity measure $m$ then respects the twin property if $m_Z = m_X$. Magnitude fulfils this property, which follows directly from the definition of a metric space because $X$ is a set and cannot include duplicate observations. That is, for $x_0 \in X$ where $(X, d)$ is a metric space, we get that $Z = x_0 \cup X = X$ and $\text{Mag}_Z = \text{Mag}_X$.
- **Monotonicity in distances.** For $|Y| = |X| \geq 2$, when $f \colon (X, d_X) \to (Y, d_Y)$ is a bijective mapping so that no distance is decreasing and at least distance is increasing and $d_X$ and $d_Y$ are negative definite, diversity does not decrease. Formally, when $d_X(x_1, x_2) \leq d_Y(f(x_1), f(x_2))$ for all $x_1, x_2 \in X$ and $d_X(a, b) < d_Y(f(a), f(b))$ for some $a, b \in X$, a diversity measure $m$ is monotone in distances if $m_X \leq m_Y$. Magnitude fulfils this conjecture for all known examples of negative definite metric spaces [21, 36].

Given that these three essential properties hold for magnitude at every choice of $t$, they also hold for the area under the magnitude function for a space with a negative definite distance metric, whose magnitude function is necessarily continuous as demonstrated in Appendix A.2. However, none of

the alternative diversity measures defined in Appendix C.1 fulfil the twin property or the condition of being monotonic in observations as demonstrated in Appendix C.4. This can lead to an undesirable decrease in diversity when including novel observations to a space that adds information and should thus not reduce the total diversity but is similar to existing points. Indeed, given this behaviour we argue that a decrease in diversity as measured by the existing metrics can actually be misleading.

**Diversity requires basic invariances.** Elaborating on the idea that diversity measures should pass basic sanity checks, the following more basic properties are desirable for measuring the intrinsic diversity of a space as intended [22]:

- **Isometry invariance:** Diversity does not change under isometric transformations of a space. Magnitude is isometry invariant as defined and proven in Appendix A.4.
- **Symmetry:** Diversity is invariant under permutations of the input observations. Formally, let $\sigma : X \to X$ be a permutation of $X$. Then, a diversity measure $m$ is symmetric if $m(X) = m(\sigma(X))$.
- **Absence invariance:** Diversity only depends on the samples and features present in the dataset. Formally, let $X' \subseteq \mathbb{R}^{D'}$ be dataset with feature space of dimension $D'$, and $X \subseteq X'$ be the subset of observed samples and non-zero features $X \subseteq \mathbb{R}^D$. Then, a diversity measure $m$ is absence-invariant if $m(X) = m(X')$. That is, diversity does not change when removing elements or features that have not been observed or have zero probability.

Magnitude and thus MAGAREA fulfil the aforementioned conditions by definition. Specifically, magnitude is absence-invariant, symmetric and isometry-invariant because the distance metric $d_X$ itself is absence-invariant, symmetric and isometry-invariant. These properties can be proven as in Appendix A.4. Overall, invariances are key properties when studying the diversity of latent representations. Indeed, symmetry is so essential that all diversity measures evaluated in this study fulfil it. GMSTDS however, is not absence invariant as it always equals zero whenever one feature or dimension in the embedding space is constant, which limits its usefulness and makes it sensitive to the absence of information.

**Diversity measures the effective number of distinct observations.** Originating from using entropy to define a suitable notion of diversity for theoretical ecology, the following aspects of measuring diversity in deserve to be highlighted:

- **Effective size:** A dataset with a fixed number of points is more diverse when points are separated e.g. distributed uniformly or maximally disordered and becomes less diverse as observations cluster together. Diversity is maximised when points are completely distinct and minimised when all observations are identical. Formally, let $e : X \to X'$ be a transformation that decreases the entropy of space $X$.[7] We then require that $m(X) \leq m(e(X))$. Say a diversity measure $m(X) \in \mathbb{R}$ has a minimum $m_{\min}$ and maximum value $m_{\max}$. We further require that $\lim_{t \to 0} m(tX) = m_{\min}$ and $\lim_{t \to \infty} m(tX) = m_{\max}$.
- **Effective number:** Diversity should be measured the effective size of a space in the range $[1, |X|] \ni m(X)$, so diversity is expressed as the effective number of distinct points or clusters in a space.

We note that the latter condition on diversity being measured as effective number is very helpful for measuring biodiversity [8] and to give a sensible summary of the clustering structure of a dataset, but being interpretable as a number of points is potentially not necessary in all applications. Nevertheless, we find that relating diversity to assessing entropy and requiring that diversity is aware of clusterability and uniformity in the data, is essential for a useful diversity measure as further illustrated in Appendix C.5.1. Note that magnitude fulfils the conditions of measuring both an effective size and an effective number by definition as defined in Section 3.2 and discussed in more detail by Leinster [22]. Based on this, MAGAREA as a multi-scale summary of magnitude also fulfils the condition of measuring the effective size of a space and can easily be converted into an effective number.

---

[7]We leave the precise definition of entropy up to discussion [22], but give examples of what we consider to be entropy-decreasing operations. One example is scaling through a negative definite metric space $tX$ by increasing the similarity between observations i.e. by decreasing $t$. Removing informative features, mode dropping or mode collapse are further phenomena that decrease diversity.

Table S.1: **Counterexamples** demonstrating that alternative diversity measures fail to fulfil fundamental axioms of diversity, whereas magnitude passes these sanity checks.

| Space | MAGAREA ($\uparrow$) | VS ($\uparrow$) | AVGSIM ($\downarrow$) | GMSTDS ($\uparrow$) |
|---|---|---|---|---|
| X two point space | 4.602 | 1.867 | 0.368 | 0.500 |
| Q absence | 4.602 | 1.867 | 0.368 | 0 |
| Z duplicate point | 4.602 | 1.77 | 0.579 | 0.471 |
| Y three point space | 4.613 | 1.809 | 0.577 | 0.469 |

**Diversity is a multi-scale summary of (dis)similarities.** Finally, we note that diversity should be seen as a continuous function of the scale of similarity and behave accordingly:

- **Similarity-sensitivity:** Diversity is computed from and determined by the (dis)similarities between observations. That is, a diversity measure $m \in \mathbb{R}$ of a dataset $X$ is defined as $m := f(D_X)$ or $m(X) := f(\zeta_X)$ for some function $f$, where $D_X$ is a matrix of distances between observations in $X$ and similarly $\zeta_X$ is a matrix of similarities.
- **Scale-dependence:** Further, we require that a diversity measure $m_t \in \mathbb{R}$ is a continuous function of the scale of (dis)similarity $t$. Thus, diversity is not just a one-number summary, but a function of said scale. Formally, $m_t(X) := f(D_X(t))$ or $m_t(X) := f(\zeta_X(t))$.
- **Multi-scale:** A multi-scale measure encodes both local and global trends in the data manifold by considering multiple levels of scale or resolution simultaneously. Let $m_t$ be a scale-dependent diversity measure. A multi-scale measure, $m$, further summarises diversity across multiple scales i.e. $m = f(m_{t_1}, (m_{t_2}, ..., (m_{t_n})) \in \mathbb{R}$ for $n > 2$ and some summary function $f$.

Rather than giving a snapshot of diversity at a fixed degree of (dis)similarity, multi-scale methods summarise diversity across varying scales of (dis)similarity. We reason that this property is advantageous to capture a more complete picture on how both coarse and more nuanced dissimilarities in observations affect diversity. Being a multi-scale summary is a distinguishing characteristic of our proposed diversity measure, MAGAREA. Alternative diversity measures, such as average similarity, the Vendi Score or magnitude computed at one scale, do not fulfil this criterion as they are single resolution snapshots computed from a fixed similarity matrix.

## C.4 Counterexamples

In the following, we now demonstrate how the diversity measures introduced in Appendix C.1 fail at some of the fundamental axioms of diversity introduced in Appendix C.3 via simple examples. Consider the following feature matrices:

$$X = \begin{bmatrix} 1 \\ 0 \end{bmatrix}, \quad Q = \begin{bmatrix} 1 & 0 \\ 0 & 0 \end{bmatrix}, \quad Z = \begin{bmatrix} 1 \\ 0 \\ 0 \end{bmatrix} \quad \text{and} \quad Y = \begin{bmatrix} 1 \\ 0 \\ 0.01 \end{bmatrix} \tag{10}$$

First, consider the two point space as given by $X$ as a reference space. We then use Manhattan distances $d_X$ and the similarity matrix $\zeta_X$ as given in Definition 3.1 to compute each diversity measure. In particular, to compute MAGAREA we use the convergence scale of $X$ as a reference. The resulting diversity values are summarised in Table S.1.

**Absence invariance.** To check for absence invariance, we include a constant feature dimension to $X$ and get $Q$. Clearly, $Q$ has the same diversity as $X$. Indeed, all diversity values are equal for these spaces other than GMSTDS. This counterexample thus shows that GMSTDS is not absence invariant.

**Twin property.** Next, we include a duplicate observation to $X$ and get $Z$ to examine the twin property. Note that including a repeated observation does not change diversity as the space still consists of only two unique observations. However, VS, AVGSIM and GMSTDS all assess that $Z$ is less diverse than $X$ and thus do not fulfil the twin property.

**Monotonicity in observations.** Lastly, include one new observation to $X$ and consider the three point space $Y$. While $Y$ is very similar to $X$, we can see that overall, in terms of absolute diversity, $X$ is not more diverse than $Y$. However, MAGAREA is the only measure in Table S.1 that indicates

Table S.2: **MAGAREA shows the correct order in diversity** when comparing the simulated examples in Figure 1a) from the main text. In contrast, two baseline diversity measures, AVGSIM and GMSTDS, as well as the discrepancy measure L2STAR fail to distinguish that the random point pattern, $X_1$, is more diverse than the clustered point pattern, $X_2$.

| | MAGAREA ($\uparrow$) | VS ($\uparrow$) | AVGSIM ($\downarrow$) | GMSTDS ($\uparrow$) | L2STAR ($\downarrow$) |
|---|---|---|---|---|---|
| $X_1$ Poisson Process | 133 | 14.6 | 0.39 | 0.59 | 0.004 |
| $X_2$ Hawkes Process | 99 | 13.1 | 0.39 | 0.59 | 0.004 |
| $X_3$ Two Gaussians | 69 | 5.7 | 0.51 | 0.53 | 0.041 |
| $X_4$ One Gaussian | 48 | 3.1 | 0.79 | 0.14 | 0.260 |

that $Y$ is slightly more diverse than $X$. Indeed, all of VS, AVGSIM and GMSTDS indicate that $Y$ is less diverse than $X$ and they are thus not monotone in observations.

**New or duplicate observations.** Including a duplicate observation should lead to lower diversity than including a new unseen observation to a space. However, GMSTDS actually indicates that $Z$, the space with duplicate points, is less diverse than the three point space $Y$, which is clearly counterintuitive.

## C.5  Simulation Studies

### C.5.1  Effective Size and Uniformity

To gain more insights, we further compare diversity metrics on the examples from Figure 1a) and report them in Table S.2. That is, we simply ask the question, which of these simulated examples are more diverse. Note that each space has the same number of points, but the examples vary in their effective size and clustering behaviour. We know a ground truth ordering, namely that the uniform pattern in $X_1$ is the most diverse example as points are most clearly separated and more evenly spread across the entire sampling domain.

Results then show that two of the baseline measures, AVGSIM and GMSTDS, fail to capture notable differences in diversity on simple simulations as they do not detect that the random pattern in $X_1$ is more diverse than a clustered pattern in $X_2$. Here, the difference in diversity between $X_1$ and $X_2$ is driven by the difference in the disorder or clustering behaviour of their respective observations. Linking back to the axioms of diversity, these examples illustrate how diversity should be aware of an effective number of distinct clusters or points and evaluate diversity via assessing entropy.

Our results thus demonstrate how magnitude gives a superior notion of diversity, that captures variations in effective size, clustering behaviour and uniformity. In contrast, both AVGSIM and GMSTDS are unaware of these differences. In practice, this could further lead to misleading assessments of diversity for e.g. generative model evaluation. For example, a clustered sentence embedding, corresponding to a few distinct groups of sentences that share the same meaning, might be wrongly deemed to have the same diversity as a more varied set of generated sentences, which notably differ in meaning and show a more uniform distribution in the embedding space.

To further contrast diversity measures against uniformity criteria, we also compute the L2-star discrepancy of each sample, L2STAR. Discrepancy measures in general assess how much an empirical distribution deviates from a uniform distribution on the unit hypercube [42, 47]. Lower discrepancy implies higher agreement with uniformity, which in our context corresponds to higher diversity. However, we observe that that L2STAR does not clearly distinguish a difference between $X_1$ and $X_2$, but rather assesses that both examples are close to uniformity. Results indicate that this classical discrepancy measure gives a global summary of evenness, which assesses $X_2$ appears uniform on a large-scale, but does not pick up on local difference in effective size and small-scale clustering behaviour. This illustrates the necessity of evaluating both local and global trends in diversity via a multi-scale summary of diversity as given by MAGAREA.

### C.5.2  Investigating the Twin Property for Diversity Measures

To link our investigation to the theoretical axioms of diversity, we examine the twin property. This requirement asserts that diversity should not change when including duplicate observations into a

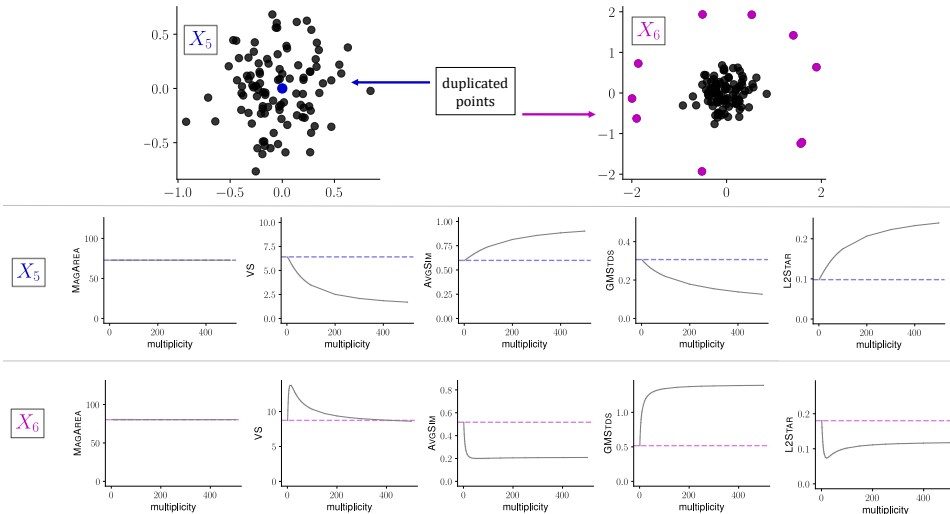

Figure S.5: **Our method, MAGAREA, is the only diversity measure that satisfies the twin property, one of the fundamental axioms of diversity. MAGAREA correctly assesses that diversity does not change when including duplicate observations.** All baseline diversity scores fail to fulfil this property and show inconsistent changes. $VS$ even switches trends from increasing initially to decreasing after a certain number of replicates has been reached for $X_6$. We investigate two simulations: $X_5$ shows a Gaussian distribution with 100 points, whose centre gets duplicated. $X_6$ shows a Gaussian blob with 100 observations as well as 10 outliers, which get duplicated with the specified multiplicity. Changes in the diversity scores as increasing numbers of duplicate copies are added are shown below. Dashed horizontal lines show the ground truth value of each diversity measure prior to including duplicate points.

given dataset. When evaluating generative models, diversity measures that satisfy the twin property are advantageous because they penalise models that just repeat existing observations again, as opposed to providing genuinely 'novel' outputs. Results of this case study are reported in Figure S.5, showing how the popular baseline diversity measures, AVGSIM, VS and GMSTDS as well as the discrepancy measure L2STAR, the L2-star discrepancy, all fail to fulfil the twin property, instead exhibiting highly-inconsistent behaviour. Our proposed method meanwhile is the only diversity measure that respects the twin property and remains consistent, demonstrating one of its practical advantages.

## D    Additional Details for Our Experiments

In the following we give further details and elaborate on the experimental setup and datasets used for our experiments as well as showcase extended results.

### D.1    Curvature Experiments

Here we provide more details about the curvature experiments, which builds on the approach by Turkes et al. [41]; see https://github.com/renata-turkes/turkevs2022on for an implementation. We generate a unit disks $D_\kappa$ of surfaces of constant curvature $\kappa$, with 3 cases: the first one is when $\kappa = 0$ (we then have the Euclidean plane), $\kappa < 0$ (we have a space of negative curvature, the Poincaré disk model of the hyperbolic plane), $\kappa > 0$ (sphere with radius $1/\sqrt{\kappa}$). We vary the curvature $\kappa$ to be in the interval $[-2, 2]$. For each value of $\kappa$, we construct point clouds by sampling 500 points from $D_\kappa$. We generate 201 surfaces with equally spaced curvature in the interval $[-2, 2]$. Then, we compute magnitude for each space using Euclidean distance and 30 evenly spaced intervals until the scale $t_{\text{cut}} = 73$.

From the resulting values of MAGAREA plotted against the curvature values in Figure 2, we can intuitively explain the relationship between diversity i.e. magnitude and curvature. For unit disks of positive curvature, the higher the curvature the lower the value of MAGAREA. This indicates that

points move closer and closer the more curved the surface is decreasing the diversity in Euclidean space. For surfaces with negative curvature we see the opposite trend. The more negatively curved the Poincaré disk the lower the value of MAGAREA. This is because Euclidean distances between points and thus diversity are decreasing.

For the results reported in Table 1 we further apply 5-fold cross-validation and aim to predict the curvature values from the diversity score. We first train a quantile regression model on the MAGAREA after applying polynomial feature transformation of degree 2 to the training data suspecting a quadratic-looking relationship between MAGAREA and curvature after exploratory analysis. Further, we compare this to piecwise linear regression with two breakpoints under the assumption that the relationship between MAGAREA and curvature as plotted in Figure 2 rather depicts a piecewise linear relationship clearly separating spaces of positive and negative curvature. Both regression models were compared in the final results in Table 1 to investigate multiple proposals on how to interpolate between the MAGAREA scores for surfaces of negative and positive curvature.

We further report six alternative models from Turkes et al. [41], which are using features from persistent homology (PH) summarising persistence diagrams (PDs). See Bubenik et al. [4] for a more detailed explanation on PH and its relationship to curvature. Specifically, in Table 1 we reproduce the following models from Figure 4. and Table 3. of Turkes et al. [41]:

- SVR (all PH features) referred to as 0-dim PH simple by Turkes et al. [41], which uses the lifespans of the persistence diagram computed on the samples;
- SVR (selected PH features) denoted 0-dim PH simple 10 by Turkes et al. [41], which uses the 10 longest lifespans; and
- SVR (PH vectorisation) corresponding to 0-dim PH by Turkes et al. [41], which selects the best PD vectorisation amongst a number of options, namely persistence images (PI) or persistence landscapes (PL).

All the PH-based methods use support vector regression (SVR) with a RBF kernel. Hyperparameter tuning for these models is conducted as reported by Turkes et al. [41] using grid search with a choice of C parameters in $\{0.001, 1, 100\}$. We further reproduce 1 method based on pairwise distance matrices:

- SVR (distance matrices) denoted as ML by Turkes et al. [41].

Finally, we restate the performance scores of these two methods directly from Turkes et al. [41]:

- MLP (shallow) denoted as NN shallow by Turkes et al. [41]; and
- MLP (deep) denoted as NN deep by Turkes et al. [41].

We also note that the other models achieve different performance scores on our dataset than reported by Turkes et al. [41] due to a slight difference in dataset and cross-validation splits. We use a smaller subset of samples than Turkes et al. [41] each having a unique curvature value as described above, and ensure that all models are evaluated on the same splits of data across 5-fold CV for fair comparison. Finally, we summarise the MSE achieved by each model in Table 1. Illustrating this, Figure 2 further shows examples of both magnitude functions for negative and positive curvature as well as the clear piecewise-linear trend between MAGAREA and curvature.

### D.2 Measuring the Intrinsic Diversity of Text Embeddings

We analyse data from Tevet and Berant [38], consisting of 1K sets of 10 sentences each generated for unique input prompts for 3 sentence generation tasks. The code by Tevet and Berant [38] is available at `https://github.com/GuyTevet/diversity-eval` under an MIT licence and data can be downloaded from `http://diversity-eval.s3-us-west-2.amazonaws.com/data.zip`. These tasks are story completion (storyGen) and dialogue response generation (respGen), both using MASS model fine-tuned on each dataset, and 3-word prompt completion (promptGen) using GPT-2-large without fine tuning. From the 1K response sets per task, 10 have been generated using the same decoding parameter, the softmax-temperature dec, sampled evenly across the range $[0.2, 1.2]$, which controls the trade-off between quality and diversity by skewing models towards avoiding low-probability tokens as dec decreases. This leads to potentially higher quality or fidelity but lower diversity or creativity in generated text. Further, for 200 response sets per task, mean human evaluation scores of text diversity were collected. Human workers rated the level of diversity

of the response set on a scale from 5 (highest) to 1 (lowest). The mean of these scores, absolute HDS (ABSHDS), measures the human perception of text diversity.

We then create embeddings for each set of sentences utilise the sentence-transformer library by Reimers and Gurevych [30]. In particular, we apply the following five embedding models:

- all-distilroberta-v1: general-purpose model, embedding dimension 768;
- all-MiniLM-L6-v2: general-purpose model, embedding dimension 384;
- all-mpnet-base-v2: general-purpose model, embedding dimension 768;
- bert-large-nli-stsb-mean-tokens: general-purpose model, embedding dimension 1024; and
- roberta-base-nli-mean-tokens: general-purpose model, embedding dimension 768.

For each text embedding we compute magnitude functions and our diversity measure, MAGAREA, across 20 evenly sampled scales in $[0, t_{\text{cut}}]$ where $t_{\text{cut}}$ is the median of the convergence scales across all embeddings setting $\epsilon = 0.01|X|$. We then run prediction tasks as described in Section 4.2 repeating the same procedure for predicting both human scores and decoding parameters. Figure S.10 details the results for predicting human scores. Figure S.11 then shows summaries of the ranks achieved by each metric across the two experiments. For additional context on the choice of convergence scales, Figure S.12 shows that changing the value of $\epsilon$ for computing MAGAREA has a small effect on its correspondence with human evaluation scores indicating that our measure is relatively robust to this choice. Results from Figure 3 are further expanded is expanded in Table S.3 which shows the mean $R^2$ scores and median ranks across datasets and Table S.4 summarising the difference in performance scores between MAGAREA and other diversity measures highlighting that entropy-based diversity metrics give superior notions of diversity. Illustrating the relationship between magnitude and diversity, Figure 4 and Figure S.7 plot both magnitude functions and the values of MAGAREA against both human scores and decoding parameters for one of the embedding models (plots for other embeddings follow similar trends). Indeed, these visualisations then demonstrate that MAGAREA can be used as a descriptor of diversity in generated text achieving high rank correlation with known ground truth values of diversity.

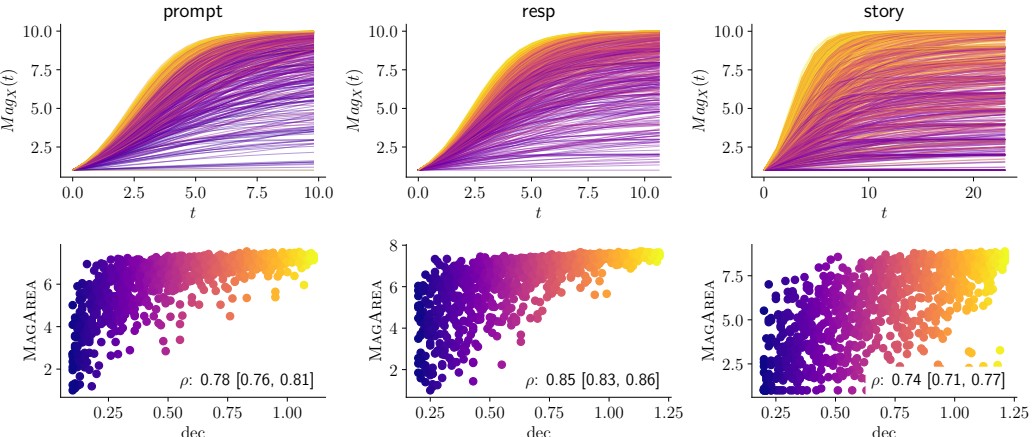

Figure S.6: **MAGAREA and magnitude functions** computed using all-mpnet-base-v2 plotted against the decoding parameters. The value $\rho$ shows the mean rank correlation between MAGAREA and the softmax-temperature as well as 95% bootstrap intervals computed by resampling 1000 times.

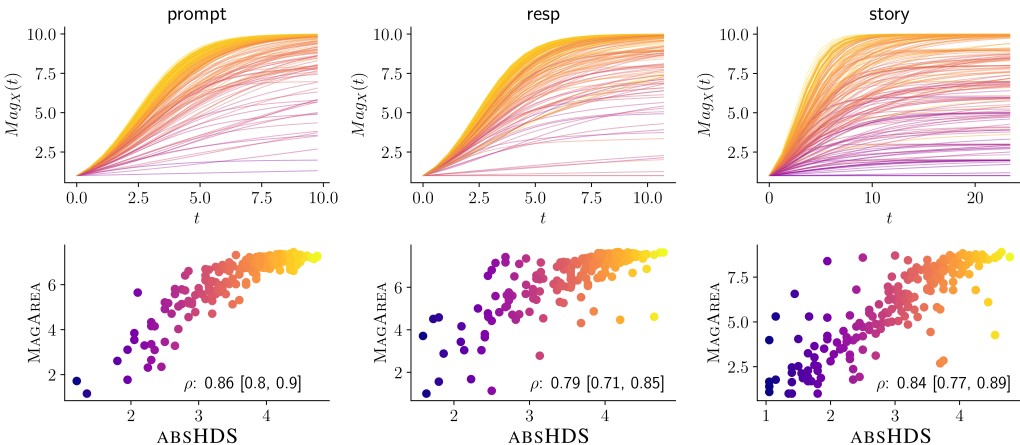

Figure S.7: **MAGAREA and magnitude functions** computed using all-mpnet-base-v2 embeddings plotted against the human scores. The value $\rho$ shows the mean rank correlation between MAGAREA and ABSHDS as well as 95% bootstrap intervals computed by resampling 1000 times.

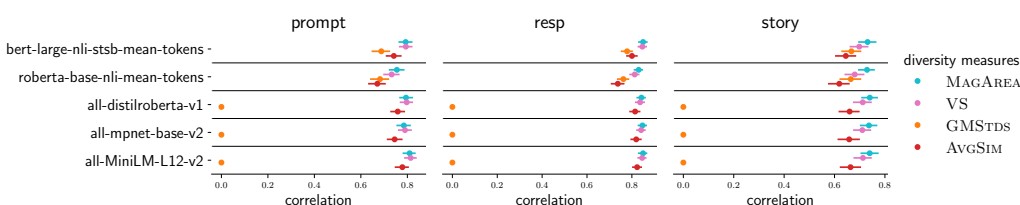

Figure S.8: **MAGAREA shows higher rank correlation with the ground truth softmax temperature than alternative diversity measures** across 3 tasks and 5 embedding models. Baseline measures, AVGSIM and GMSTDS, show noticeably worse rank correlation. Points show the mean Spearman correlation and lines represent 95% bootstrap intervals computed across 1000 resamples.

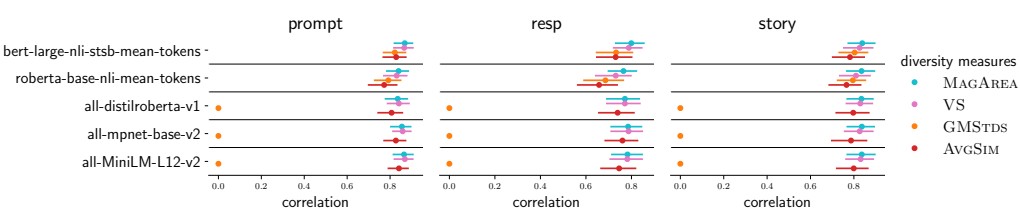

Figure S.9: **MAGAREA shows higher rank correlation with human evaluation scores than alternative diversity measures** across 3 tasks and 5 embedding models. Baseline measures, AVGSIM and GMSTDS, show noticeably worse rank correlation. Points show the mean Spearman correlation and lines represent 95% bootstrap intervals computed across 1000 resamples.

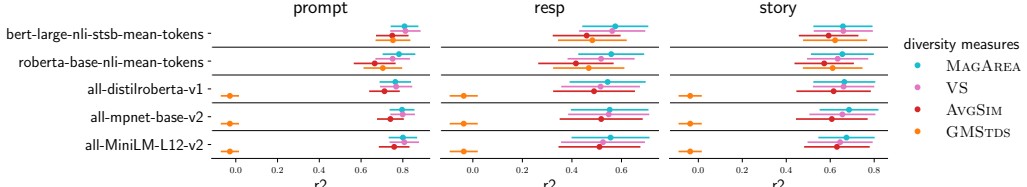

Figure S.10: **MAGAREA predicts human evaluation scores**, across 3 tasks and 5 embedding models. Baseline measures, AVGSIM and GMSTDS, perform worse in terms of the mean $R^2$ scores. Points show the mean of the $R^2$ scores, while lines represent the standard deviations across 5-fold cross-validation (repeated 10 times).

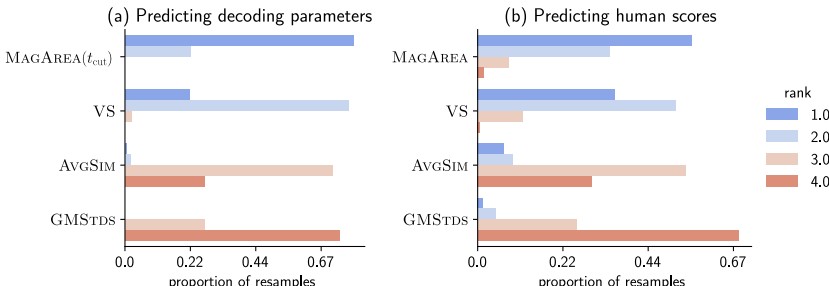

Figure S.11: **The area under the magnitude function outranks baseline diversity measures** at (a) predicting decoding parameters and (b) predicting human-evaluated diversity scores across all experiments and cross-validation resamples.

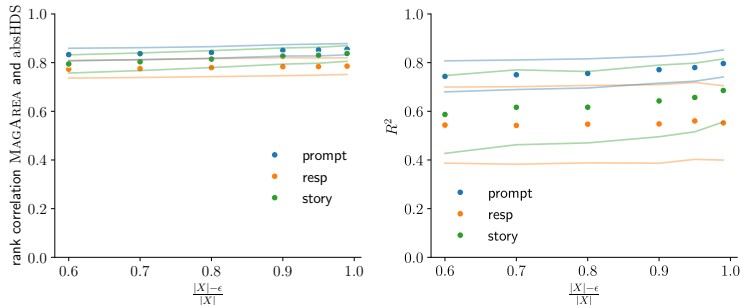

Figure S.12: **Impact of the choice of convergence scales on predicting human evaluation scores**. Both the rank correlation between MAGAREA and absHDS as well as the $R^2$ scores achieved by MAGAREA for predicting absHDS remains consistent across different choices of convergence scales. Here we take the median convergence scale across embeddings as a cutoff scale, but change the $\epsilon$ parameter from Definition 3.5. In the left plot, points show the mean rank correlation across 1000 repeated bootstrap resampling and lines the corresponding standard deviations. The right plot, points show the mean performance scores with standard deviations from 20 repeated 5-fold CV.

Table S.3: **The mean performance of each diversity measure** in terms of $R^2$ scores for predicting the decoding parameter. We also report 95% percentile intervals of these scores as well as standard deviations.

| Model | Median Rank | Mean $R^2$ | Standard Deviation | Lower 95% PI | Upper 95% PI | Task |
|---|---|---|---|---|---|---|
| MAGAREA | 1 | 0.62 | 0.05 | 0.50 | 0.70 | Prompt |
| VS | 2 | 0.61 | 0.06 | 0.49 | 0.71 | Prompt |
| AVGSIM | 3 | 0.55 | 0.06 | 0.42 | 0.66 | Prompt |
| GMSTDS | 4 | 0.19 | 0.24 | −0.02 | 0.55 | Prompt |
| MAGAREA | 1 | 0.70 | 0.04 | 0.62 | 0.77 | Resp |
| VS | 2 | 0.69 | 0.04 | 0.60 | 0.76 | Resp |
| AVGSIM | 3 | 0.64 | 0.07 | 0.50 | 0.74 | Resp |
| GMSTDS | 4 | 0.23 | 0.30 | −0.02 | 0.66 | Resp |
| MAGAREA | 1 | 0.53 | 0.05 | 0.44 | 0.62 | Story |
| VS | 2 | 0.49 | 0.06 | 0.38 | 0.58 | Story |
| AVGSIM | 3 | 0.41 | 0.06 | 0.29 | 0.51 | Story |
| GMSTDS | 4 | 0.17 | 0.22 | −0.03 | 0.50 | Story |

Table S.4: **The difference between each diversity measure and MAGAREA** in terms of the difference in $R^2$ scores when predicting the decoding parameter. We also report 95% percentile intervals of these differences and standard deviations.

| Measure | Mean Difference in $R^2$ Scores | Standard Deviation | Lower 95% PI | Upper 95% PI | Dataset |
|---|---|---|---|---|---|
| VS | 0.00 | 0.02 | −0.02 | 0.06 | Prompt |
| AVGSIM | 0.07 | 0.03 | 0.03 | 0.15 | Prompt |
| GMStds | 0.42 | 0.26 | 0.05 | 0.71 | Prompt |
| VS | 0.01 | 0.01 | −0.01 | 0.04 | Resp |
| AVGSIM | 0.07 | 0.05 | 0.00 | 0.17 | Resp |
| GMStds | 0.47 | 0.30 | 0.07 | 0.77 | Resp |
| VS | 0.04 | 0.02 | 0.01 | 0.09 | Story |
| AVGSIM | 0.12 | 0.03 | 0.06 | 0.18 | Story |
| GMStds | 0.36 | 0.23 | 0.06 | 0.62 | Story |

### D.3 Characterising Text Embedding Spaces

For further experiments, we analyse 16384 embedded documents of from four different HuggingFace datasets as processed by Wayland et al. [43]. In particular, we use the following datasets:

- arXiv: abstracts of all arXiv articles up to the end of 2021;
- bbc: summaries of BBC news articles;
- cnn: summaries of news articles from CNN and DailyMail; and
- patents: abstracts of U.S. patent applications.

The first $2^{14}$ samples from the corresponding training datasets were embedded using the sentence-transformer library by Reimers and Gurevych [30]. The following pre-trained models were used:

- all-distilroberta-v1: general-purpose model, embedding dimension 768;
- all-MiniLM-L6-v2: general-purpose model, embedding dimension 384;
- all-mpnet-base-v2: general-purpose model, embedding dimension 768; and
- multi-qa-distilbert-cos-v1: QA-specialized model, embedding dimension 768, maximum sequence length 512 word pieces.

Further, embeddings from two large language models, ada-002 (embedding dimension: 1 536) and mistral-embed (embedding dimension: 1 024) were obtained through queries via the corresponding APIs of their providers (OpenAI and MistralAI, respectively).

We then use PCA to reduce each embedding space to 384 dimensions to obtain a comparable dimensionality. This is done to mitigate some of the influence the difference in dimensionalities can have on the results of the analysis. Further, we sample 300 points at random from each space, repeating this procedure 200 times, which yields a dataset of 1000 embeddings generated by different models for each dataset. We further compute MAGAREA across 20 scales up to the median convergence point across all embeddings per dataset again using cosine distances and setting $\epsilon = 0.05|X|$. Similarly, we take the pairwise magnitude differences MAGDIFF between all subsample's magnitude functions. This is compare it to alternative diversity measures, VS, GMSTDS and AVGSIM computed as defined in Appendix C.1. For each dataset we then use a simple 5-NN classifier to classify the embedding model based on the one number summaries such as MAGAREA and report the mean and standard deviation of the accuracy across 5-fold cross-validation with 20 repetitions. We compare this to the more expressive descriptor MAGDIFF, which we similarly use as a pre-computed distance matrix for 5-NN classification. Figure S.13 illustrates the pairwise magnitude differences between all subsamples for each dataset. Table 2 then shows the results of this classification task. For further sensitivity analysis, Table S.5 shows analogous results that were computed across varying choices of $k$ neighbours for $k$-NN classification. Results demonstrate that classification accuracy remains consistent across varying parameter choices.

Table S.5: **Classification performance remains consistent across varying choices of $k$ for $k$-NN classification.** We show the accuracy ($\uparrow$) of different diversity scores for distinguishing between six embedding models of the bbc dataset, using PCA pre-processing and a $k$-NN classifier across varying values of $k$. These results are analogous to Table 2 in the main text.

| k | MAGDIFF | AVGSIM | VS | GMSTDS |
|---|---|---|---|---|
| 1 | **0.93 ± 0.01** | 0.84 ± 0.02 | 0.72 ± 0.03 | 0.66 ± 0.03 |
| 3 | **0.94 ± 0.01** | 0.84 ± 0.02 | 0.72 ± 0.02 | 0.66 ± 0.03 |
| 5 | **0.95 ± 0.01** | 0.84 ± 0.02 | 0.72 ± 0.03 | 0.66 ± 0.03 |
| 7 | **0.95 ± 0.01** | 0.84 ± 0.02 | 0.73 ± 0.02 | 0.66 ± 0.03 |
| 9 | **0.95 ± 0.01** | 0.84 ± 0.02 | 0.74 ± 0.02 | 0.66 ± 0.03 |
| 11 | **0.95 ± 0.01** | 0.84 ± 0.02 | 0.74 ± 0.02 | 0.66 ± 0.03 |

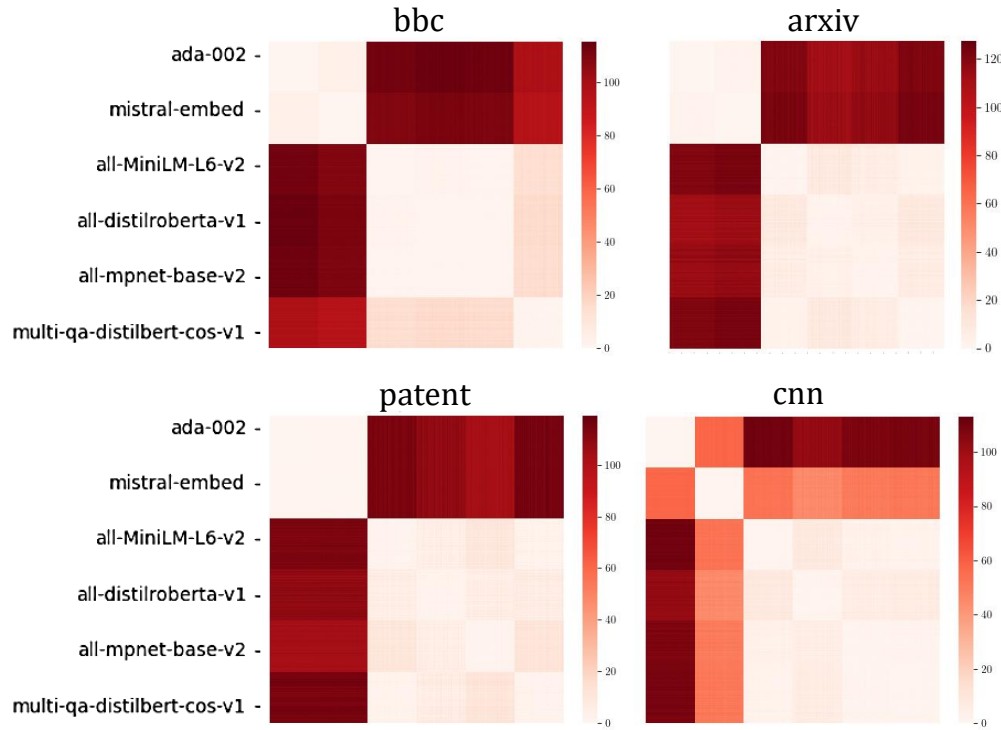

Figure S.13: **Pairwise MAGDIFF for subsamples from six different embedding models and four document datasets.** For each dataset, magnitude is computed until the median convergence scale. The plot shows the absolute MAGDIFF normalised by this cut-off scale to show the average absolute difference in magnitude across the evaluation interval. We see a clear block-wise structure between the pairwise magnitude differences of each subsample highlighting that the magnitude functions of these samples are clearly distinct between different embedding models.

## D.4  Image Experiments

We adapt code by Friedman and Dieng [13] to download and process the CIFAR10 test dataset using torchvision. Specifically, we use utility functions from https://github.com/vertaix/Vendi-Score available under an MIT licence. Further, these data are embedded using Inception_v3 [37]. We then simulate sequential and simultaneous mode dropping for this embeddings. We compute magnitude for 10 scales sampled uniformly from $t_0 = 0$ until the 95% convergence scale of the reference embedding where each class is still evenly represented. The main results in Figure 5 then report the relative changes in MAGDIFF across the evaluation interval. In Figure S.14 we further report the precision and density for the same simulated mode dropping scenarios as explained in the image embedding experiments. We also show the relative decrease in magnitude computed at three fixed scales at 33%, 50% and 100% of the convergence scale of the reference space chosen to explore a variation of resolutions. Finally, Figure S.15 summarises analogous results for different variations of the Vendi Score, which has also been proposed to measure mode dropping. We observe that while VS shows steady trends when computed using a linear kernel or the Laplacian kernel on normalised embeddings. However, VS does not perform well at when simply using the Laplacian kernel without preprocessing, which demonstrates that finding the right scale of similarity is also important for VS and it is possible to misspecify the degree of similarity between observations leading to unreliable performance.

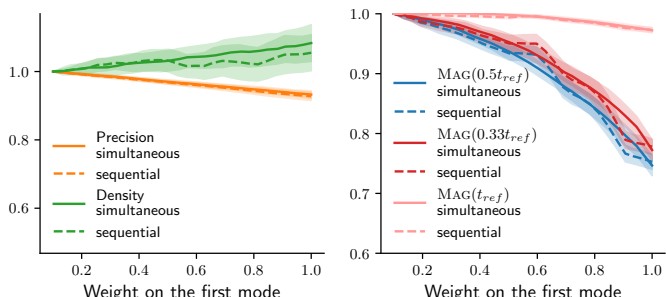

Figure S.14: Line plots of the proportion of points on the first mode against recall, coverage, relative difference in magnitude at $t = 0.5$ and magnitude function difference relative to the reference across simultaneous sampling vs. sequential sampling. Lines show the mean values of each normalised metric across 20 resamples, shaded areas the standard deviations.

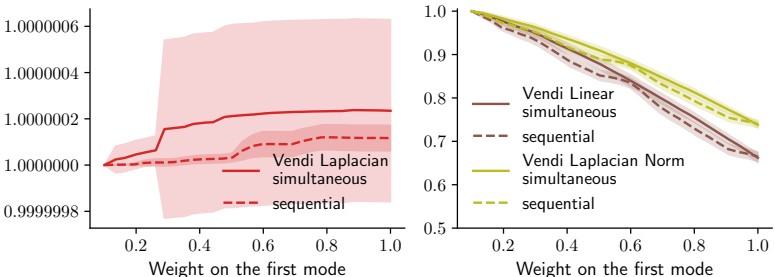

Figure S.15: Line plots of the proportion of points on the first mode against the relative change in different variations of the Vendi Score, VS, across two different mode dropping scenarios. VS is computed using a Laplacian kernel without (left) and with normalisation (right) as well as a linear kernel (right). Lines show the mean values of each normalised metric across 20 resamples, shaded areas the standard deviations.

### D.5 Graph Embedding Experiments

To assess our new diversity measure's utility for graph generative model evaluation we reproduce the benchmark by Thompson et al. [40] and include our proposed reference-based diversity measure, MAGDIFF, to the diversity evaluation benchmark. The code for reproducing this graph evaluation benchmark is available at `https://github.com/uoguelph-mlrg/GGM-metrics` under an MIT licence. Specifically, we conduct this experiment on five graph datasets:

- **Lobster:** A dataset consisting of 100 stochastic graphs generated so that each node is at most 2 hops removed from a backbone path and the number of vertices varies between 10 and 100.[7, 40]
- **Grid:** A dataset of 100 two-dimensional graphs consisting of 100 to 400 vertices [7, 24, 40, 46].
- **Proteins:** A dataset of 918 protein networks. Each vertex is an amino acid and edges connect amino acids that are less than 6 Angstroms away from each other [9]. Only graphs with between 100 to 500 vertices are selected [7, 24, 40, 46].
- **Ego:** A dataset of 757 graphs that are 3-hop networks with 50 to 399 vertices [40, 46]. These graphs were extracted from the CiteSeer citation network where nodes represent documents [34].
- **Community:** A dataset with 500 two-community graphs with between 60 to 160 vertices, where each community has been generated using the Erdős-Rényi model [11] setting $n$ equal to half the number of vertices and $p = 0.3$. Additional edges amounting to 5% of the number of vertices have been added to each graph with uniform probability [40, 46].

Further, this experiment uses a Graph Isomorphism Network [45, GIN] architectures as an embedding model and following the procedure by [40] we vary the following hyperparameters for these models:

We vary the number of layers between $[2, 3, \ldots, 7]$ and vary the hidden dimensions in the interval $[5, 30]$ with an increment of 5 resulting in a total of 36 architectures. We repeat the experiments for 5 different random seeds. The experimental setup used to evaluate the evaluation metrics for both mode collapse and mode dropping then is as follows: First $P_r \approx P_g$, so that $P_r$, the real distribution, is identical to $P_g$, the generated distribution. Then the perturbation parameter $p \in [0, 1]$ is introduced, which transforms the generated graph datasets step-wise and increases the dissimilarity (and hence diversity) between the reference and generated datasets. Therefore, we use it as a proxy to measure the difference in diversity between $P_r$ and $P_g$. To evaluate this decrease in diversity, we compute magnitude for the corresponding graph embeddings across 40 evenly-spaced scales until the convergence scale of the reference choosing $\epsilon = 0.05|X|$. For precision, recall, density and coverage we take the parameter $k = 5$, as proposed previously by [28], to ensure a fair comparison. We then normalise all metrics such that their value is 0 when $P_r = P_g$ (which is exactly when the degree of permutation is 0). For this, we follow the normalisation strategy by [40] and normalise MAGDIFF by the cardinality of each embedding. Next, we vary the parameter $p$ and compute each evaluation metric. We report the Spearman correlation coefficient between each metric and the degree of the perturbation $p$. Hence, the value of a metric which captures the decrease in diversity accurately should increase with the increase of $p$, and rank correlation of 1 corresponds to an ideal metric. Results for the whole experiment across all datasets are presented in Figure S.16. The violin and boxplots reported in this figure then summarise the distribution of each evaluation measures rank correlation to the degree of perturbation across the 5 random seeds and the aforementioned hyperparameter choices influencing the embedding models. Finally, Figure S.17 investigates the influence the choice of convergence scale has on the results of these experiments and we observe that low values of $\epsilon$ lead to better agreement with the true degree of perturbation. Further, the trends in the value of MAGDIFF are stable across choices of $\epsilon \leq 0.05|X|$ as chosen throughout this study.

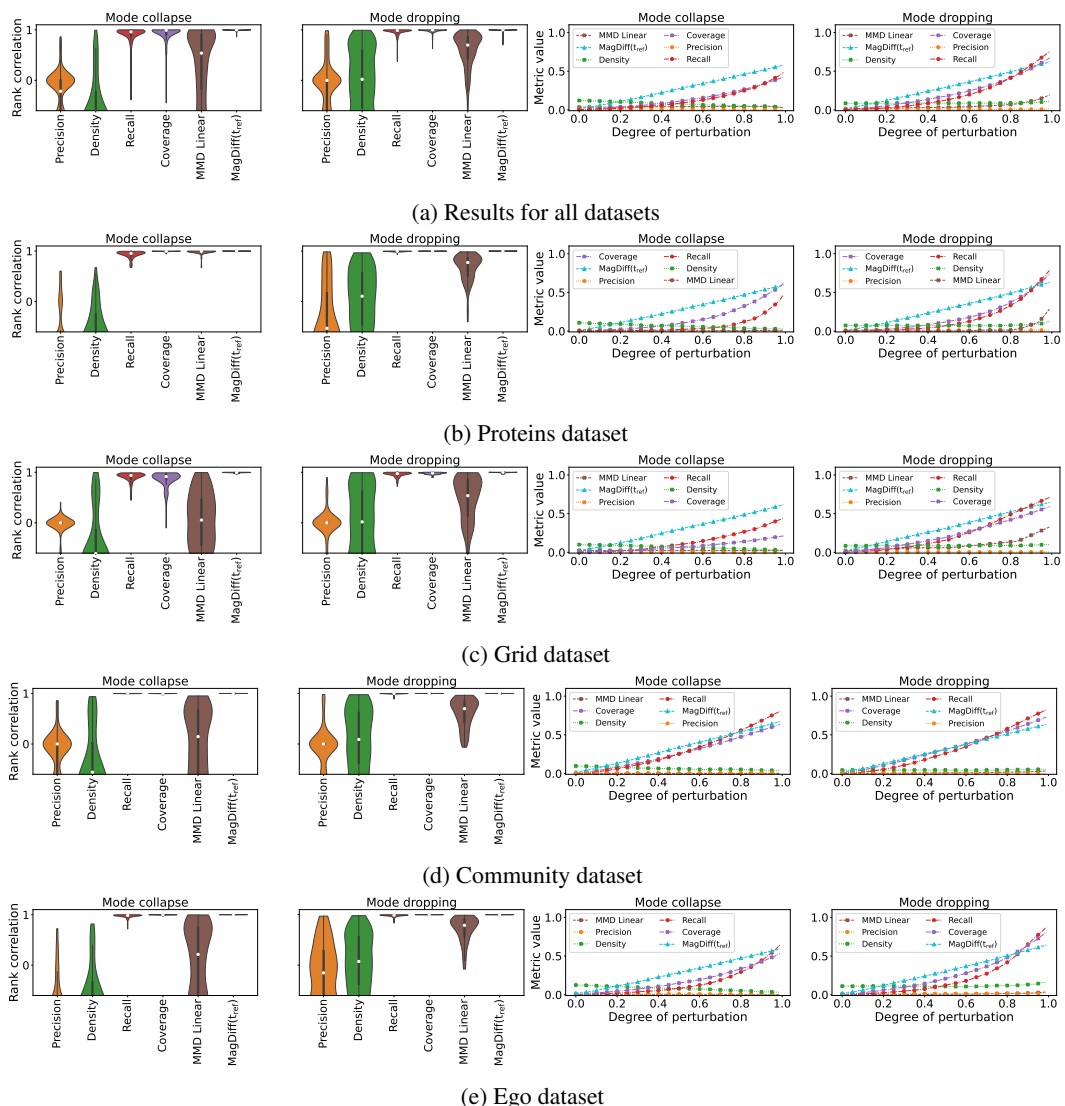

(a) Results for all datasets

(b) Proteins dataset

(c) Grid dataset

(d) Community dataset

(e) Ego dataset

Figure S.16: **Results for the mode collapse and mode dropping experiments**. The patterns for each of the datasets is similar to the results on the Lobster graphs, which we show in the paper.

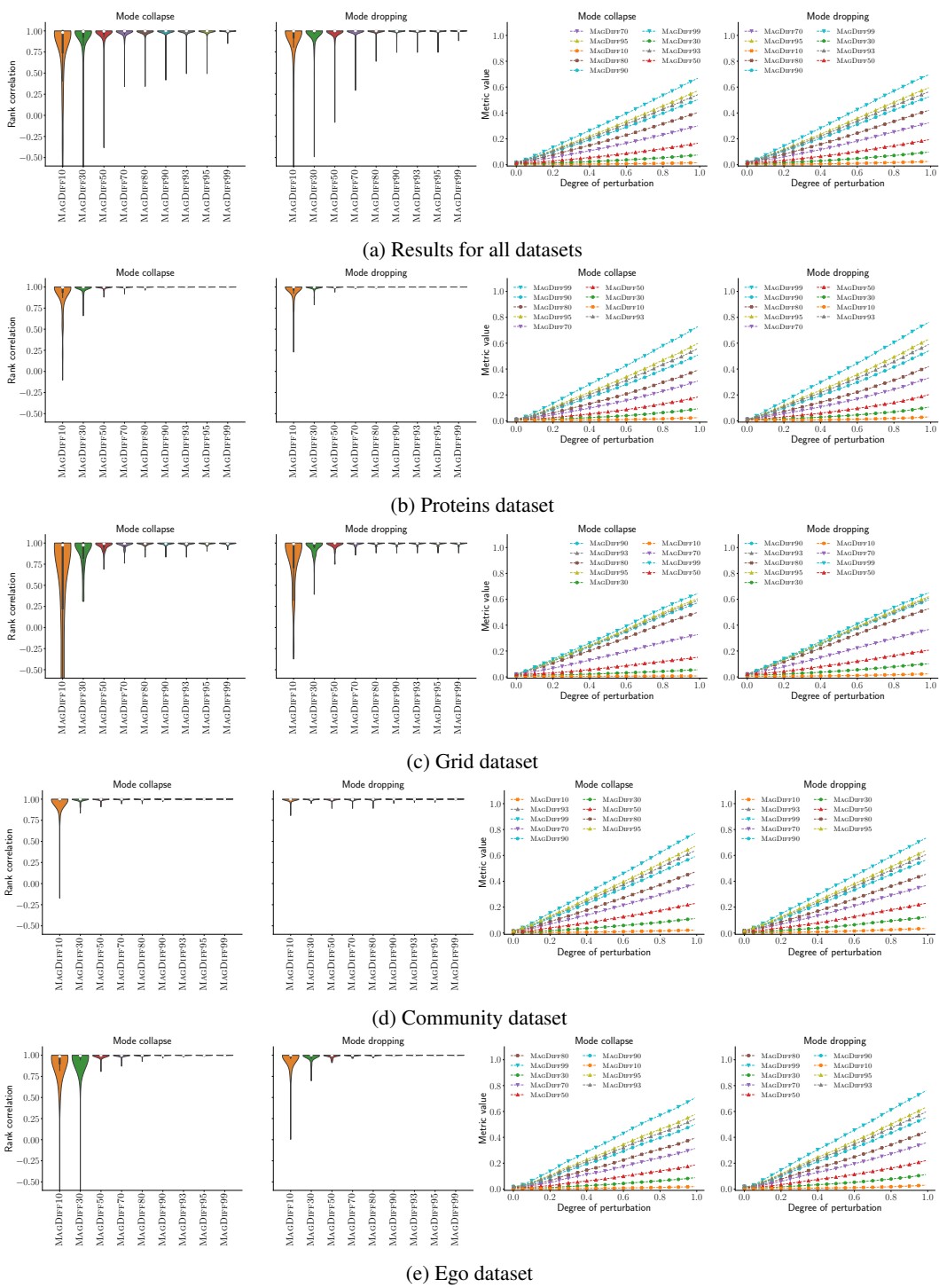

(a) Results for all datasets

(b) Proteins dataset

(c) Grid dataset

(d) Community dataset

(e) Ego dataset

Figure S.17: **Rank correlation between MAGDIFF and the degree of perturbation for different choices of convergence scale.** Here we vary the choice of $\epsilon$ influencing the reference scale that is chosen to compute $\text{MAGDIFF}(|X|-\epsilon/|X|)$ for the mode collapse and mode dropping experiments. We clearly observe that low values of $\epsilon$ as given by MAGDIFF95 or MAGDIFF99 lead to higher rank correlation and better agreement with the true decrease in diversity.

