# OpenReview forum: "Metric Space Magnitude for Evaluating the Diversity of Latent Representations"
_NeurIPS.cc/2024/Conference — NeurIPS 2024 poster_

### Official Review · Reviewer_WoGi · 2024-07-10

**Soundness:** 3
**Presentation:** 3
**Contribution:** 3
**Rating:** 7
**Confidence:** 4

**Summary:**

This paper studies the diversity in representation learning. The authors propose a novel family of diversity measures based on metric space magnitude, a mathematical invariant that captures numerous important multi-scale geometric characteristics of metric spaces.

The main contributions are as follows:
1. The authors first introduce magnitude as a general tool for evaluating the diversity of latent representations,
2. The authors formalise a notion of difference between the magnitude of two spaces across multiple scales of similarity.

**Strengths:**

1. The idea of introducing magnitude as a general tool for evaluating the diversity of latent representations is quite attractive and it seems can be generalized.
2. The authors formalise a notion of difference between the magnitude of two spaces across multiple scales of similarity.
3. The authors discuss the theoretical properties a suitable diversity measure should satisfy.

**Weaknesses:**

1. As mentioned in the paper, it is difficult to define the diversity. The paper does not clearly explain how to link their method with diversity. Maybe I did't get their point. But I think their method is more like to evaluate the better representation instead of measuring the diversity?
2. The authors demonstrate that magnitude is stable and can detect curvature, however such connection makes people a little confused.

**Questions:**

See weakness.
1. Can authors better explain my confuses? I think the proposed approach is more like to evaluate the better representation instead of measuring the diversity.
2. I didn't see clearly connections between magnitude and curvature. Can authors explain clearly?

**Limitations:**

See weaknesses.

---

> ### Author Rebuttal · Authors · 2024-08-06
>
> We thank the reviewer for their questions and the positive assessment of our work.
>
> **For revisions, we take this feedback as motivation to clarify and further explain the links between magnitude and diversity.**
>
> ___
>
>
> > As mentioned in the paper, it is difficult to define the diversity. The paper does not clearly explain how to link their method with diversity. Maybe I didn't get their point. But I think their method is more like to evaluate the better representation instead of measuring the diversity?
>
> Thanks for raising this question about our current write-up! We will use this to revise our manuscript. To summarise our main points:
>
> In our paper, we define diversity by considering fundamental axioms which a sensible diversity measure should satisfy, namely monotonicity in distance (diversity should increase as dissimilarity between observations increases) and the twin property (diversity should not change when including duplicate observations). We state these main theoretical requirements, and demonstrate that our proposed measures fulfil them. In this way, we provide the theoretical link between diversity and our methods.
>
> Specifically, we use the magnitude of a latent representation to evaluate diversity. Magnitude measures diversity as the “effective number of points” at a distance scale or “zoom factor.” Given a distance between data points, and a zoom factor, magnitude answers the question: How many *distinct* points or clusters can be distinguished at this scale?
>
> Intuitively, diverse spaces have clearly separated observations and high magnitude. In comparison, less diverse spaces, whose observations are more clustered, have lower magnitude. We then define formalised measures of diversity that summarise trends in magnitude across multiple resolutions.
>
> This is further illustrated in Figure 1 in the paper. Here, the random pattern in $X_1$ is the most diverse example because its points are most clearly separated and more evenly spread across the entire area. Thus, it achieves the highest values of magnitude. In comparison, the clustered pattern in $X_2$ generated via a Hawkes process is less diverse because it shows a more uneven distribution with points being concentrated at specific areas. Overall, the magnitude of $X_2$ is thus lower than for $X_1$. Our diversity evaluation pipeline illustrated in Figure 1 then summarises these differences in diversity across multiple resolutions in a principled manner.
>
> In the context of representation learning, diversity then is a key concept on what it means for representations to be better. Our goal thus is to evaluate latent representations by measuring their diversity. **In revisions, we will revise the introduction of magnitude and diversity to better explain what diversity means in our context and how our method measures diversity. Further, we will add an extended case study to the appendix in order to illustrate how our method measures diversity via intuitive examples.**
>
>
> ___
>
>
>
> > I didn't see clearly connections between magnitude and curvature. Can authors explain clearly?
>
> We thank the reviewer for this question. The motivation of the curvature experiment was to substantiate our argument that magnitude encodes important geometrical properties with experimental evidence. Our results then support and are motivated by the theoretical results linking magnitude and curvature ([Willerton 2010](https://arxiv.org/pdf/1005.4041)).
>
> **At the same time, this experiment shows that magnitude is more effective at this task than other, more complex geometrical summaries** (like persistent homology).
>
> For the experiment itself, we can intuitively explain the relationship between diversity i.e. magnitude and curvature as illustrated in Figure S.5. For unit disks of positive curvature, the higher the curvature the lower the value of MagArea. This indicates that points move closer and closer the more curved the surface is decreasing the diversity in Euclidean space.
>
> For surfaces with negative curvature we see the opposite trend. The more negatively curved the Poincaré disk the lower the value of MagArea. This is because Euclidean distances between points and thus diversity are decreasing.
>
> In revisions, we will clarify the connection between magnitude, diversity and curvature. To do so, we will reference the relationship to relevant theoretical results in the text. Further, we will add an introductory text earlier in the paper to motivate this experiment.

---

> > ### Comment · Reviewer_WoGi · 2024-08-10
> >
> > Thank you for your clarifications and additional materials. It makes me understand it clearly.

---

> > > ### Author Response · Authors · 2024-08-12
> > >
> > > Thank you very much for your positive response. We are pleased to hear the additional materials and clarifications have provided a clearer understanding.
> > >
> > > Should you or any other reviewers have additional questions, we are happy to discuss and provide further clarifications.

---

> > > > ### Comment · Area_Chair_cagw · 2024-08-12
> > > > **Could you answer some further questions?**
> > > >
> > > > Dear Authors,
> > > >
> > > > Thank you for taking the time to discuss this with the Reviewers.
> > > >
> > > > At the request of the Program Chair, I would like to take this opportunity to suggest that you, as Area Chair, respond to the Reviewer's question in more detail, as I believe that the Authors' response was unclear and that this will hinder my writing of a meta-review.
> > > >
> > > > The Reviewer originally asked about how the method linked to diversity. I understand that Authors answered the question partially, but I would appreciate it if Authors could answer the following:
> > > >
> > > > - What are the formal definitions of the four axioms (Monotonicity, Twin property, Absence invariant, Multi-scale)? The first three are somewhat trivial, but how to define the multi-scale i.e., "encodes both local and global trends in the data manifold." mathematically does not seem trivial.
> > > > - Does the proposed metric MAGAREA satisfy the four axioms? If so, where in the paper can we find the proof?
> > > > - Is the design of MAGAREA unique? Cannot we replace $\\exp$ in the definition of $\\zeta$ with another increasing and convex function?
> > > >
> > > > Also, as a relevant question, I would like to ask why we cannot use for authors' motivation (traditional) discrepancy measures in the low-discrepancy sequence area, like the star-discrepancy (e.g., [WS08]. We can find traditional discussions in e.g., [War72])?
> > > >
> > > > [War72] Warnock, Tony T. "Computational investigations of low-discrepancy point sets." Applications of number theory to numerical analysis. Academic Press, 1972. 319-343.
> > > > [WS08] Xiaoqun Wang, Ian H. Sloan,
> > > > Low discrepancy sequences in high dimensions: How well are their projections distributed?, Journal of Computational and Applied Mathematics, Volume 213, Issue 2,
> > > > 2008, Pages 366-386,
> > > >
> > > > Thank you in advance for taking the time for the discussion.

---

> ### Author Response · Authors · 2024-08-13
> **Response 1/3**
>
> We thank the AC and the reviewers for their further questions and are happy about the chance to elaborate more on the technical details regarding the link between the mathematical theory of diversity and our proposed method. Our previous response aimed to convey an intuitive understanding of diversity.
>
>
> > What are the formal definitions of the four axioms (Monotonicity, Twin property, Absence invariant, Multi-scale)? The first three are somewhat trivial, but how to define the multi-scale i.e., "encodes both local and global trends in the data manifold." mathematically does not seem trivial.
>
> > Does the proposed metric MAGAREA satisfy the four axioms? If so, where in the paper can we find the proof?
>
> Yes, MAGAREA fulfils all the desired axioms as we will detail below. Definitions of the diversity axioms are stated in Appendix C.3. and we are happy to elaborate on this discussion for revisions.
>
> First, we want to highlight what it means for a measure to be multi-scale. A **multi-scale measure** encodes both local and global trends in the data manifold by considering multiple levels of scale or resolution simultaneously. Formally, we can require that a diversity measure $m_t \in \mathbb{R}$ is a continuous function of the scale of dissimilarity $t$. A multi-scale measure, $m$, then summarises diversity across multiple scales i.e. $m = f(m_{t_1}, (m_{t_2}, …, (m_{t_n})) \in \mathbb{R}$ for $n>2$ and some summary function $f$. That is, rather than giving a snapshot of diversity at a fixed degree of (dis)similarity, multi-scale methods summarise diversity across varying scales of (dis)similarity.
>
> We reason that this property is advantageous to capture a more complete picture on how both coarse and more nuanced dissimilarities in observations affect diversity. Indeed, **being a multi-scale summary is a distinguishing characteristic of our proposed diversity measure, MAGAREA.** Alternative diversity measures, such as average similarity, the Vendi score or magnitude computed at one scale, do not fulfil this criterion as they are single resolution snapshots computed from a fixed similarity matrix. **For revisions, we will clarify the current statement in the main text and include a formal definition as well as an extended discussion regarding this property in the appendix.**
>
> Regarding the remaining properties, we can prove that monotonicity in observations, the twin property and absence invariance hold for the magnitude of a negative definite metric space. Then, because magnitude $Mag_X(t)$ fulfils these axioms for each scale of $t \in \mathbb{R}^+$, our measure $\text{MAGAREA} = \int_{T} Mag_X(t) dt$ does as well. We state this in Appendix C.3. and **will elaborate on the relevant proofs during revisions.**
>
> Briefly, we can sketch the relevant proofs as follows:
> - Magnitude is **monotone in observations** i.e. magnitude does not decrease when including a novel observation. This follows from [Corollary 2.4. in Leinster (2013)](https://arxiv.org/pdf/1012.5857).
> - Magnitude fulfils the **twin property** i.e. it remains unchanged when including a duplicate observation. This follows from the fact that a metric space cannot contain a duplicate element by definition.
> - A diversity measure is **absence-invariant** if it remains unchanged when restricting the empirical distribution to its support. That is, diversity does not change when removing elements or features that have not been observed or have zero probability. Magnitude is absence-invariant because the distance matrix $d$ itself is absence-invariant. **We will add this proof during revisions.**
>
> We thank the AC for their questions and will highlight the relevant diversity axioms, their definitions and proofs better in revisions. Please let us know whether additional clarifications are required.

---

> ### Author Response · Authors · 2024-08-13
> **Response 2/3**
>
> > Is the design of MAGAREA unique? Cannot we replace $\exp$ in the definition of $\zeta$ with another increasing and convex function?
>
> The computation of MAGAREA is unique insofar as we require $\zeta(t)=f(-td)$ being invertible for all $t \in  \mathbb{R}^+$, which, to the best of our understanding, won't necessarily hold for any  increasing and convex function $f$. For example, take $f(x)=x^{1000}$, so that $\zeta_{ij}(t) = (t d_{ij})^{1000}$ which is increasing and convex for $td > 0$. Then, for some low values of $t$ close to zero, $\zeta$ will be singular and hence not invertible.
>
> We know, however, that any positive definite matrix is invertible so that we can define the magnitude of any such matrix $\zeta$ [(Leinster 2017)](https://arxiv.org/pdf/1606.00095). Given this motivation, $\exp$ is a somewhat “canonical” choice for defining the magnitude of a metric space because it is a prime example of a strictly positive definite kernel, which ensures invertibility of the similarity matrix for any negative definite metric $d$ [(Feragen et al. 2015)](https://ieeexplore.ieee.org/document/7298922).
>
> From a category-theoretic perspective, magnitude represents a generalised notion of size for a metric space, which is a special type of monoidal categories. The exponential kernel then has been chosen for defining the magnitude of a metric space because of its multiplicative properties  i.e. it is necessary that $f(x+y)=f(x) \cdot f(y)$ [(Leinster 2021, p.212)](https://arxiv.org/pdf/2012.02113) to define a valid notion of size. This essentially forces, $f(x) = c^{-x}$ for some constant $c$. Hence, the choice of $e$ as a basis is arbitrary and any other positive constant could be used, which is equivalent to re-scaling the distances.
>
> We also want to note that the current definition of magnitude has revealed useful theoretical and geometric insights, such as proving the connection between magnitude and the curvature of a metric space. Hence, if we choose any $f$ that does not fulfil the multiplicative property above, we lose the existing knowledge relating magnitude to geometrical properties of the underlying space, for example.
>
> The standard formulation of $\zeta$ also has some appealing properties linking it to diversity. We have $\exp(-td_{ii})=\exp(0)=1$, so that the similarity of an observation to itself is always $1$. Further, we have $\lim_{t \to \infty} \exp(-t  d_{ij}) = 0$, so that the similarity between distant points approaches zero asymptotically. Thus, $\zeta(t)$ gives a valid similarity matrix, whose entries are bounded by $[0,1]$, which is desirable for defining a (dis)similarity-based notion of diversity [(Leinster 2021, p.173)](https://arxiv.org/pdf/2012.02113). **In fact this behaviour is the practical reason why magnitude can be interpreted as the effective number of points.**
>
> Hence, we know that other choices of $\zeta$ are possible and we believe it is worth investigating the generalisation of our methods to other choices of kernels in future work. Nevertheless, we find that, for now, $\exp$ provides a useful “default” choice for defining diversity based on the considerations detailed above.

---

> > ### Author Response · Authors · 2024-08-13
> > **Response 3/3**
> >
> > > Also, as a relevant question, I would like to ask why we cannot use for authors' motivation (traditional) discrepancy measures in the low-discrepancy sequence area, like the star-discrepancy (e.g., [WS08]. We can find traditional discussions in e.g., [War72])?
> >
> > Thank you for pointing out these references! We agree that discrepancy measures, such as star-discrepancy, give an interesting set of tools for quantifying the difference between a latent space and a uniform distribution on a hypercube. In our context, these methods could be used to measure an important aspect of diversity, namely the evenness or uniformity of an empirical distribution. In our previous response, we explained the importance of evenness for understanding diversity via the comparison between the clustered patterns $X_2$ and the random point pattern $X_1$.
> >
> > Preliminary experiments with discrepancy measures on such data show that (a) measures are often not capable of reproducing the ground-truth ranking in diversity, i.e. $X_1, X_2, X_3, X_4$, or (b) fail to capture the degree of difference in diversity when comparing $X_1$ and $X_2$. **We will incorporate these measures in an extended experimental discussion in our revision.**
> >
> > Moreover, we find that “evenness” does not fully capture all relevant aspects of diversity. Another key aspect is measuring the absolute richness of a dataset. For example, we want to measure that a space with 100 uniformly distributed samples does not decrease in diversity when including additional 10 randomly sampled points. That is, we define diversity not just by the evenness of the empirical distribution, but also by the richness or the number of distinct observations or clusters. Linking back to the axioms of diversity, this behaviour is described by requiring monotonicity in observations as well as the twin property. We can show that both axioms do not hold for the star-discrepancy via counterexamples. For example, when including discrepancy measures into the simulation study conducted for the twin property as detailed in the additional PDF, we see that L2-star discrepancy changes under the inclusion of duplicates.
> >
> > In comparison, our proposed diversity measure evaluates both evenness and richness by summarising the effective number of distinct points across multiple resolutions. Therefore, our method gives a unique view on the diversity of latent spaces that is based on mathematical theory, but not yet addressed by existing diversity or discrepancy measures in ML.
> >
> > Regarding the original motivation of our work, we focused on linking our method to two applications that benefit from improved diversity evaluation, namely the evaluation of generative models and automated embedding-based diversity evaluation e.g. for assessing LLMs. To the best of our knowledge, discrepancy measures have not yet been included as standard benchmark methods for diversity evaluation in these fields. **We fully agree that this is worth further investigation and are excited to include discrepancy measures as alternative baselines during revisions.**
> >
> > We thank the AC for their questions as well as their further suggestions and are happy to clarify our answers.

---

> > > ### Comment · Area_Chair_cagw · 2024-08-14
> > > **Thank you for your response!**
> > >
> > > Dear Authors, thank you for your response. We will consider your response in the discussion between Reviewer and AC.

---

### Official Review · Reviewer_cXLQ · 2024-07-10

**Soundness:** 3
**Presentation:** 3
**Contribution:** 3
**Rating:** 6
**Confidence:** 4

**Summary:**

This paper focuses on evaluating the diversity of latent representations. The authors develop a family of magnitude-based measures of the intrinsic diversity of latent representations, formalizing a novel notion of dissimilarity between magnitude functions of finite metric spaces. Moreover, they demonstrate the practicality and performance of the proposed measures in different domains and tasks.

**Strengths:**

1. The method proposed in this paper is innovative, and the writing is logical.
2. This paper conducts an in-depth theoretical analysis and sufficient experimental discussion and analysis.

**Weaknesses:**

1. In Section 4.3, the reason for choosing the 5-NN classifier needs to be explained. In addition, what is the purpose of designing a comparative experiment between PCA pre-processing and no pre-processing?
2. This paper discusses several application scenarios of the proposed diversity evaluation, and we can further explore whether there are more scenarios worth exploring, such as measuring the representation ability of graph contrastive learning models.

**Questions:**

1. In Table 1, what is the reason for the apparent performance difference between MAGAREA with the piecewise linear and MAGAREA using quantile regression?
2. In section 4.3, what is the basis for the author to use the 5-NN classifier to predict the embedding model?
3. In section 4.3, what is the purpose of the author's comparison of the results of PCA pre-processing and no pre-processing? How do the authors view that these two methods have both positive and negative effects on different models?
4. In Section 4.5, the authors discussed magnitude evaluation graph generative models. Can the diversity evaluation proposed in this paper also be used to evaluate the representation learning ability of graph contrastive learning models?

**Limitations:**

1. MAGDIFF is a reference-free measure of intrinsic diversity, but does not measure fidelity.
2. The paper does not investigate whether embedding-based similarities are outperformed by alternative task- or domain-specific similarities. Instead, the evaluation relies on the utility of embedding models and assumes that latent spaces encode useful/realistic relationships between samples.

---

> ### Author Rebuttal · Authors · 2024-08-06
>
> We thank the reviewer for their feedback and are looking forward to discussing the questions raised as well as clarify them in the text during revisions.
>
> ___
>
> > In section 4.3, what is the basis for the author to use the 5-NN classifier to predict the embedding model?
>
> The 5-NN classifier is chosen as a very simple model, which somewhat surprisingly already manages to separate the embedding spaces of different models extremely well based on their difference in intrinsic diversity. Here, 5 neighbours are chosen as a default as implemented in the $\texttt{sklearn}$ package without any hyperparameter tuning. Motivated by your question, we conducted further sensitivity analysis to assess how much the reported results change under different parameter choices. Table 2 in the attached PDF then shows that classification accuracy hardly varies across different choices of $k$ neighbours. For revisions, we will add an explanation for the choice of model in the text.
>
> ___
>
> > In section 4.3, what is the purpose of the author's comparison of the results of PCA pre-processing and no pre-processing? How do the authors view that these two methods have both positive and negative effects on different models?
>
> PCA preprocessing is applied under the assumption that the varying dimensionality between embedding spaces could be the main driver of the observed disparities in diversity. We thus wanted to test if dimensionality reduction to the same number of 384 dimensions could account for the differences in diversity and reduce the classifiers’ performance scores. However, our results show that this does not change the experimental results. There thus remain inherent differences in each models’ signature as distinguished via diversity. We will include further explanations and motivations for these model choices in the revisions. **We also want to clarify that our method does not depend on such a specific pre-processing; we merely used it here to make the task of detecting a specific model harder.**
>
> ___
>
>
> > In Table 1, what is the reason for the apparent performance difference between MAGAREA with the piecewise linear and MAGAREA using quantile regression?
>
> Thank you for pointing this out! We will clarify this in revisions: The input data to these models, that is the values of MagArea for different curvature values, is plotted in Figure S.5. Explanatory analysis of this relationship then informed our model choice. The piecewise linear model better fits the trend in Figure S.5, which is why it outperforms the quadratic relationship modelled via quantile regression. Both models were included to offer multiple proposals on how to interpolate between the MagArea scores for surfaces of negative and positive curvature.
>
> ___
>
> > In Section 4.5, the authors discussed magnitude evaluation graph generative models. Can the diversity evaluation proposed in this paper also be used to evaluate the representation learning ability of graph contrastive learning models?
>
> We appreciate this suggestion and are excited to explore and discuss more scenarios for which our proposed method is of relevance. Evaluating representation learning models themselves is an exciting question in itself that deserves further exploration.
>
> In the **context of graph contrastive learning**, we have reasons to believe that our method can be extended to the evaluation of self-supervised representations. As an intrinsic measure of the diversity of a space, magnitude measures the effective size of an embedding space. Conceptually, this can be likened to measuring the effective rank of an representations as computed by [RankMe (Garrido et al. 2023)](https://arxiv.org/pdf/2210.02885) for the evaluation of joint-embedding self supervised learning. That is, we believe that problems such as **dimensional collapse** could also be assessed in an expressive manner by a representation’s multiscale magnitude.
>
> Further, it is of interest to explore the role of diversity in a self-supervised context and investigate how diversity measures can be effectively used to improve model performance, potentially via **maximising diversity during training**.
>
> We thus believe that using magnitude for evaluating representation models deserves its **own thorough investigation** as well as further theoretical links to learning theory in SSL, which we look forward to conducting in the future.

---

### Official Review · Reviewer_vep5 · 2024-07-16

**Soundness:** 3
**Presentation:** 3
**Contribution:** 3
**Rating:** 6
**Confidence:** 3

**Summary:**

In this paper, authors introduce magnitude of a metric space to evaluate the diversity of the learned latent representation.

Extensive experimental results are proposed, which to some extent illustrate the effectiveness of the proposed criterion.

**Strengths:**

1. The expression of the paper is easy to follow.

2. The solving problem is important and could have good impact in the community.

3. The proposed method has shown good theoretical and empirical performance to some extent, illustrating the effectiveness of the proposed measurement.

4. The experimental results are comprehensive.

**Weaknesses:**

1. The theoretical analysis lacks comparison.

2. More discussion should be conducted over the circumstances when the proposed algorithm performs not as good as the existing algorithms to give more understanding of the pros and cons of the proposed criterion.

3. It is welcomed if a case study can be conducted to illustrate more insight of the proposed criterion.

**Questions:**

1. More theoretical comparison between existing methods will make the results more convincing.  Is the proposed method superior over other counterparts theoretically?

2. Can the proposed criterion combine with the existing methods to provide more comprehensive description of the learned representation and come up with better performance. Is it compatible with the existing methods?

**Limitations:**

See the questions and weakness of the paper.

---

> ### Author Rebuttal · Authors · 2024-08-06
>
> > The theoretical analysis lacks comparison.
>
> > More theoretical comparison between existing methods will make the results more convincing. Is the proposed method superior over other counterparts theoretically?
>
> We appreciate the interest in discussing theoretical analyses in comparison to existing algorithms and briefly summarise our results (we will further improve this in our revisions): Section 3.1 and Appendix C.3 discuss the theoretical properties and fundamental axioms of diversity in comparison to existing baseline measures of intrinsic diversity. Further, Appendix C.4 demonstrates which diversity axioms hold for which of these measures. For revisions, we extend this investigation of theoretic properties to a simulation study reported in Figure 1 of the attached PDF. Our magnitude-based diversity score is the **only approach** that fulfils all desired criteria, showing  that our method is superior to alternative diversity measures from a theoretical perspective. We will highlight this theoretical discussion in revisions and emphasise the importance of fulfilling fundamental axioms of diversity via an extended simulation study.
>
> ___
>
>
> > More discussion should be conducted over the circumstances when the proposed algorithm performs not as good as the existing algorithms to give more understanding of the pros and cons of the proposed criterion.
>
> **Throughout our experiments on state of the art diversity evaluation benchmarks, we have not encountered a scenario, where our methods perform worse at measuring diversity than existing methods.**
>
> In terms of the cons of our method, we noted the main limitations of our approach in Section 3.5:
> * Our method does not assess fidelity.
> * Scalability w.r.t to the size of embeddings.
>
> We note that practitioners can assess fidelity using specialised scores. Our method can then be interpreted **alongside** fidelity metrics, covering diversity aspects not yet addressed by existing measures.
>
> Further, scalability was not a limitation in our experiments. Relevant diversity evaluation tasks typically study small graph datasets, evaluate the response of text generation models for specific tasks, or study image embeddings in terms of meaningful subsets (like measuring intra-class diversity). However, in case scalability becomes an issue, we can work with efficiently approximations of our score, based on subsets.
>
> We believe that the advantages of our method by far outweigh its limitations.
> To sum up, the main benefits of our methods are:
> * Their agreement with fundamental axioms of diversity.
> * Their expressivity.
> * Their multiscale nature.
> * Their flexibility (w.r.t choosing a dissimilarity).
> * Their connection to geometric properties.
>
> We thus believe that magnitude provides a hitherto-unaddressed perspective on diversity that complements additional evaluation measures. For revisions, we will include a broader discussion on the pros and cons.
>
>
> ___
>
>
>
> > It is welcomed if a case study can be conducted to illustrate more insight of the proposed criterion.
>
> We agree it is important to build this intuition! For revisions, we provided an **extended case study in the attached PDF**, showing a comparison of our proposed magnitude-based approach and alternative measures.
>
> To link our investigation to the theoretical axioms of diversity, we examine the so-called twin property. This requirement asserts that diversity should not change when including duplicate observations into a given dataset. When evaluating generative models, diversity measures that satisfy the twin property are advantageous because they penalise models that just repeat existing observations again, as opposed to providing genuinely “novel” outputs.
>
> Results of this case study are reported in Figure 1 of the attached PDF, showing how the popular baseline measures **all fail to fulfil** the twin property, instead exhibiting  highly-inconsistent behaviour. Our proposed method meanwhile is the **only** diversity measure that respects the twin property and remains consistent, demonstrating another one of its practical advantages.
>
> To gain more insights, we further compare diversity metrics on the examples from Figure 1 and report them in Table 1 of the attached PDF. The results show that two of the baseline measures fail to capture notable differences in diversity on simple simulations as they do not detect that the random pattern in $X_1$ is more diverse than a clustered pattern in $X_2$.
>
>
> ___
>
>
>
> > Can the proposed criterion combine with the existing methods to provide more comprehensive description of the learned representation and come up with better performance. Is it compatible with the existing methods?
>
> In general, our method is compatible with existing scores. Our proposed diversity measure can be used to evaluate and improve the **performance of generative models**. This allows us to choose a generative model that best captures the diversity of a reference (achieving low MagDiff values) while simultaneously retaining high fidelity scores. Section 4.5 explores this scenario for evaluating graph generative models; we will focus on combinations of existing scores with our score in future work, thanks for the suggestion!
>
> Our measure is very versatile and can be extended to other settings to improve model performance via **incorporating it into the model training**, for instance 1) for optimisation purposes, as part of the loss function; 2) as an early stopping criterion for monitoring the diversity of the learnt representations; or 3) by improving how well the learnt representation preserves the ground truth diversity of a known reference. We would be excited to further explore these scenarios in future work.

---

> > ### Comment · Reviewer_vep5 · 2024-08-12
> > **Comment**
> >
> > Thanks for the detailed responses, especially the additional discussion, analysis and the example, which solve my concerns and questions.

---

### Author Rebuttal · Authors · 2024-08-06

We thank the reviewers for their unanimous support of our work and contributions, as well as for their interesting questions and suggestions for further clarifications. We are confident that we can implement the required changes mentioned below by making **small amendments** to our manuscript. Moreover, to complement these changes, we address reviewers’ specific questions in our responses to their individual reviews, and we are happy to provide further clarifications in the discussion phase.

---

### Decision · Program_Chairs · 2024-09-25

**Decision:**

Accept (poster)

**Comment:**

The paper successfully proposed a reference-free representation diversity metric family. Reviewers agreed that the paper tackled important problems, the paper's presentation was clear, and the idea was sufficiently supported by the experiments.

The Area Chair suggests that there might be room for improvement in justifying the proposed metric. The Authors listed four axioms for a metric to satisfy:

- **Monotonicity in observations**: Including a new observation does not decrease diversity.
- **Twin property**: Including a duplicate observation already in the set does not change diversity.
- **Absence invariant**: Diversity only depends on the samples and features present in the dataset.
- **Multi-scale**: Diversity encodes both local and global trends in the data manifold.

However, the first three are automatically satisfied if we take the sum $\\sum\_{i,j \in X} f(d(x\_i,x\_j))$ for **arbitrary** non-increasing function $f$. Also, it considers all the pairs, so it considers both *local* and *global* trends in some sense. Hence, it could be better to refine the axioms in the desiderata section. Perhaps the Authors prefer to add an additional axiom.

Nevertheless, the Area Chair judged that the above presentation issue was minor, which led to the decision this time.